# Tropical Expressivity of Neural Networks

## Abstract

We propose an algebraic geometric framework to study the expressivity of linear activation neural networks. A particular quantity that has been actively studied in the field of deep learning is the number of linear regions, which gives an estimate of the information capacity of the architecture. To study and evaluate information capacity and expressivity, we work in the setting of tropical geometry—a combinatorial and polyhedral variant of algebraic geometry—where there are known connections between tropical rational maps and feedforward neural networks. Our work builds on and expands this connection to capitalize on the rich theory of tropical geometry to characterize and study various architectural aspects of neural networks. Our contributions are threefold: we provide a novel tropical geometric approach to selecting sampling domains among linear regions; an algebraic result allowing for a guided restriction of the sampling domain for network architectures with symmetries; and an open source library to analyze neural networks as tropical Puiseux rational maps. We provide a comprehensive set of proof-of-concept numerical experiments demonstrating the breadth of neural network architectures to which tropical geometric theory can be applied to reveal insights on expressivity characteristics of a network. Our work provides the foundations for the adaptation of both theory and existing software from computational tropical geometry and symbolic computation to deep learning.

## 1 Introduction

Deep learning has become the undisputed state-of-the-art for data analysis and has wide-reaching prominence in many fields of computer science, despite still being based on a limited theoretical foundation. Developing theoretical foundations to better understand the unparalleled success of deep neural networks is one of the most active areas of research in modern statistical learning theory. *Expressivity* is one of the most important approaches to quantifiably measuring the performance of a deep neural network—such as how they are able to represent highly complex information implicitly in their weights and to generalize from data—and therefore key to understanding the success of deep learning.

*Tropical geometry* is a reinterpretation of algebraic geometry that features piecewise linear and polyhedral constructions, where combinatorics naturally comes into play [e.g., 1, 2, 3]. These characteristics of tropical geometry make it a natural framework for studying the linear regions in a neural network—an important quantity in deep learning representing the network information capacity [4, 5, 6, 7, 8, 9, 10]. The intersection of deep learning theory and tropical geometry is a relatively new area of research with great potential towards the ultimate goal of understanding how and why deep neural networks perform so well. In this paper, we propose a new perspective for measuring and estimating the expressivity and information capacity of a neural networks by developing and expanding known connections between neural networks and tropical rational functions in both theory and practice.

**Related Work.** Tropical geometry has been used to characterize deep neural networks with piece-wise linear activation functions, including two of the most popular and widely-used activation functions, namely, rectified linear units (ReLUs) and maxout units. The first explicit connection between tropical geometry and neural networks establishes that the decision boundary of a deep neural network with ReLU activation functions is a tropical rational function [11]. Concurrently, it was established that the maxout activation function fits input data by a tropical polynomial [12]. These works considered neural networks whose input domain is Euclidean, which was recently developed to incorporate tropically-motivated input domains, in particular, the tropical projective torus [13]. Most recently, tropical geometry has been used to construct convolutional neural networks that are robust to adversarial attacks via tropical decision boundaries [14].

**Contributions.** In this paper, we establish novel algebraic and geometric tools to quantify the expressivity of a neural network. Networks with a piecewise linear activation compute piecewise linear functions where the input space is divided into areas; the network computing a single linear function on each area. These areas are referred to as the *linear regions* of the network; the number of distinct linear regions is a quantifiable measure of expressivity of the network [e.g., 5]. In our work, we not only study the number of linear regions, we aim to understand their *geometry*. The main contributions of our work are the following.

- We provide a geometric characterization of the linear regions in a neural network via the input space: estimating the linear regions is typically carried out by random sampling from the input space, where randomness may cause some linear regions of a neural network to be missed and result in an inaccurate information capacity measure. We propose an *effective sampling domain* as a ball of radius $R$, which is a subset of the entire sampling space that hits all of the linear regions of a given neural network. We compute bounds for the radius $R$ based on a combinatorial invariant known as the *Hoffman constant*, which effectively gives a geometric characterization and guarantee for the linear regions of a neural network.

- We exploit geometric insight into the linear regions of a neural network to gain dramatic computational efficiency: when networks exhibit invariance under symmetry, we can restrict the sampling domain to a *fundamental domain* of the group action and thus reduce the number of samples required. We experimentally demonstrate that sampling from the fundamental domain provides an accurate estimate of the number of linear regions with a fraction of the compute requirements.

- We provide an open source library integrated into the Open Source Computer Algebra Research (OSCAR) system [15] which converts both trained and untrained arbitrary neural networks into algebraic symbolic objects. This contribution then opens the door for the extensive theory and existing software on symbolic computation and computational tropical geometry to be used to study neural networks.

The remainder of this paper is organized as follows. We provide an overview of the technical background on tropical geometry and its connection to neural networks in Section 2. We then devote a section to each of the contributions listed above—Sections 3, 4, and 5, respectively—in which we present our theoretical contributions and numerical experiments. We close the paper with a discussion on limitations of our work and directions for future research in Section 6.

## 2   Technical Background

In this section, we give basic definitions from tropical geometry required to write tropical expressions for neural networks.

### 2.1   Tropical Polynomials

Algebraic geometry studies geometric properties of solution sets of polynomial systems that can be expressed algebraically, such as their degree, dimension, and irreducible components. *Tropical geometry* is a variant of algebraic geometry where the polynomials are defined in the *tropical semiring*, $\mathbb{R} = (\mathbb{R} \cup \{\infty\}, \oplus, \odot)$ where the addition and multiplication operators are given by $a \oplus b = \max(a, b)$ and $a \odot b = a + b$, respectively. Define $a \oslash b := a - b$.

Using these operations, we can write polynomials as $\bigoplus_m a_m T^m$, where $a_i$ are coefficients, $T \in \bar{\mathbb{R}}$, and where the sum is indexed by a finite subset of $\mathbb{N}^n$. In our work, we consider the following generalizations of tropical polynomials.

**Definition 2.1.** A *tropical Puiseux polynomial* in the indeterminates $T_1, \ldots, T_n$ is a formal expression of the form $\bigoplus_m a_m T^m$ where the index $n$ runs through a finite subset of $\mathbb{Q}_{\geq 0}^m$ and $T^m = T_1^{m_1} \odot \cdots \odot T_n^{m_n}$, and taking powers in the tropical sense.

**Definition 2.2.** A *tropical Puiseux rational map* in $T_1, \ldots, T_n$ is a tropical quotient of the form $p \oslash q$ where $p, q$ are tropical Puiseux polynomials.

Tropical (Puiseux) polynomials and rational maps induce functions from $\mathbb{R}^n \to \mathbb{R}$, which take a point $x \in \mathbb{R}^n$ to the number obtained by substituting $T = x$ in the algebraic expression and performing the (tropical) operations. It is important to note that tropically, the formal algebraic expression contains strictly more information than the corresponding function, since different tropical expressions can induce the same function.

## 2.2 Tropical Expressions for Neural Networks

We now overview and recast the framework of [11], which establishes the first explicit connection between tropical geometry and neural networks, in a slightly different language for our results.

As in [11], the neural networks we will focus on are fully connected multilayer perceptrons with ReLU activation, i.e., functions $\mathbb{R}^n \to \mathbb{R}^m$ of the form $\sigma \circ L_d \circ \sigma \circ L_{i-1} \circ \cdots \circ L_1$ where $L_i : \mathbb{R}^{n_{i-1}} \to \mathbb{R}^{n_i}$ is an affine map and $\sigma(t) = \max\{t, 0\}$. For the remainder of this paper, we use the term "neural network" to refer solely to these. We will always assume that the weights and biases of our neural networks are rational numbers. From a computational perspective, this is not a serious restriction since this is sufficient to describe any neural network with weights and biases given by floating point numbers. We refer to the tuple $[n, n_1, \ldots, n_{d-1}, m]$ as the *architecture* of the neural network.

One of the key observations intersecting tropical geometry and deep learning is that, up to rescaling of rational weights to obtain integers, neural networks can be written as tropical rational functions [11, Theorem 5.2]. From a more computational perspective, it is usually preferable to avoid such rescaling and simply work with the original weights. The proof of Theorem 5.2 in [11] can directly be adapted to show that any neural network can be written as the function associated to a tropical Puiseux rational map. In their language, this corresponds to saying that any neural network is a *tropical rational signomial* with nonnegative rational exponents.

# 3 Sampling Domain Selection Using a Hoffman Constant

Estimating the number of linear regions of a neural network typically proceeds by sampling points from the input domain and counting the memberships of these points. To guarantee that membership is exhaustive, we seek a sampling domain as a sufficiently large ball so that all linear regions are intersected. At the same time, we would like for the ball to be as small as possible to guarantee efficient sampling. We are thus searching for the smallest ball from which we can sample in such a way that all linear regions are intersected. Given the polyhedral geometry of tropical Puiseux rational maps, it turns out that the radius of this smallest ball that we seek is closely related to the *Hoffman constant*, which is a combinatorial invariant.

Our contribution in this section is a definition of a Hoffman constant of a neural network; we demonstrate its relationship to the smallest sampling ball and propose algorithms to compute its true value and lower and upper bounds.

## 3.1 Defining a Neural Network Hoffman Constant

In simpler terms, the Hoffman constant can be expressed for a matrix as follows. Let $A$ be an $m \times n$ matrix. For any $b \in \mathbb{R}^m$, let $P(A, b) = \{x \in \mathbb{R}^n : Ax \leq b\}$ denote the polyhedron determined by $A$ and $b$. For a nonempty polyhedron $P(A, b)$, let $d(u, P(A, b)) = \min\{\|u - x\| : x \in P(A, b)\}$ denote the distance from a point $u \in \mathbb{R}^n$ to the polyhedron, measured under an arbitrary norm $\| \cdot \|$

on $\mathbb{R}^n$. Then there exists a constant $H(A)$ only depending on $A$ such that

$$d(u, P_{A,b}) \le H(A)\|(Au - b)_+\| \tag{1}$$

where $x_+ = \max\{x, 0\}$ is applied coordinate-wise [16]. The constant $H(A)$ is called the *Hoffman constant* of $A$.

**The Hoffman Constant for Tropical Polynomials and Rational Functions.** Let $f : \mathbb{R}^n \to \mathbb{R}$ be a tropical Puiseux polynomial and let $\mathcal{U} = \{U_1, \dots, U_m\}$ be the set of linear regions of $f$. Let $f(x) = a_{i1}x_1 + \dots + a_{in}x_n + b_i$ occur on the region $U_i$. Further, let $A = [a_{ij}]_{m \times n}$ be the matrix of coefficients in the expression of $f$ over $\mathcal{U}$. The linear region $U_i$ is defined by the following inequalities

$$a_{i1}x_1 + \dots + a_{in}x_n + b_i \ge a_{j1}x_1 + \dots + a_{jn}x_n + b_j, \quad \forall\, j = 1, 2, \cdots, m. \tag{2}$$

In matrix form, (2) is equivalent to

$$(A - \mathbf{1}a_i)x \le b_i\mathbf{1} - b \tag{3}$$

where $\mathbf{1}$ is a column vector of all 1's; $a_i$ is the $i$th row vector of $A$; and $b$ is a column vector of all $b_i$. Denote $\widetilde{A}_{U_i} := A - \mathbf{1}a_i$ and $\widetilde{b}_{U_i} := b_i\mathbf{1} - b$. Then the linear region $U_i$ is captured by the linear system of inequalities $\widetilde{A}_{U_i}x \le \widetilde{b}_{U_i}$.

**Definition 3.1.** Let $f : \mathbb{R}^n \to \mathbb{R}$ be a tropical Puiseux polynomial. The *Hoffman constant of $f$* is defined as

$$H(f) = \max_{U_i \in \mathcal{U}} H(\widetilde{A}_{U_i}).$$

Care needs to be taken in defining a Hoffman constant for a tropical Puiseux rational map: We want to avoid having all linear regions defined by systems of linear inequalities, since there exist linear regions which are not convex. To do so, we consider convex refinements of linear regions induced by intersections of linear regions of tropical polynomials.

**Definition 3.2.** Let $p \oslash q$ be a difference of two tropical Puiseux polynomials. Let $A_p$ (respectively $A_q$) be the $m_p \times n$ (respectively $m_q \times n$) matrix of coefficients for $p$ (respectively $q$). The *Hoffman constant of $p \oslash q$* is

$$H(p \oslash q) := \max \left\{ H\left( \begin{bmatrix} A_p \\ A_q \end{bmatrix} - \mathbf{1} \begin{bmatrix} a_{i_p} \\ a_{i_q} \end{bmatrix} \right) : i_p = 1, \cdots, m_p;\ i_q = 1, \cdots, m_q \right\}. \tag{4}$$

Let $f$ be a tropical Puiseux rational map. Then the *Hoffman constant of $f$* is defined as the minimal Hoffman constant of $H(p \oslash q)$ over all possible expressions of $f = p \oslash q$.

Given the correspondence between neural networks and tropical Puiseux rational maps, the Hoffman constant is well-defined for any neural network and may be computed from the geometry and combinatorics of its linear regions.

## 3.2 The Minimal Effective Radius

For a neural network whose tropical Puiseux rational map is $f : \mathbb{R}^n \to \mathbb{R}$, let $\mathcal{U} = \{U_1, \dots, U_m\}$ be the collection of all linear regions. For any $x \in \mathbb{R}^n$, define the *minimal effective radius* of $f$ at $x$ as

$$R_f(x) := \min\{r : B(x, r) \cap U_i \ne \emptyset, U_i \in \mathcal{U}\}$$

where $B(x, r)$ is the ball of radius $r$ centered at $x$. That is, $R_f(x)$ is the minimal radius such that the ball $B(x, r)$ intersects all linear regions. It is the smallest required radius of sampling around $x$ in order to express the full classifying capacity of the neural network $f$.

We start with the following lemma which relates the minimal effective radius to the Hoffman constant when $f$ is a tropical Puiseux polynomial.

**Lemma 3.3.** *Let $f$ be a tropical Puiseux polynomial and $x \in \mathbb{R}^n$ be any point, then*

$$R_f(x) \le H(f) \max_{U_i \in \mathcal{U}} \|(\widetilde{A}_{U_i}x - \widetilde{b}_{U_i})_+\|. \tag{5}$$

In particular, we are interested in studying when $\mathbb{R}^m$ and $\mathbb{R}^n$ are equipped with the $\infty$-norm. In this case, the minimal effective radius can be related to the Hoffman constant and function value of $f = p \oslash q$. For a tropical Puiseux polynomial $p(x) = \max_{1 \leq i \leq m_p}\{a_i x + b_i\}$, let $\check{p}(x) = \min_{1 \leq j \leq m_q}\{a_j x + b_j\}$ be its min-conjugate.

**Proposition 3.4.** *Let $f = p \oslash q$ be a tropical Puiseux rational map. For any $x \in \mathbb{R}^n$, we have*

$$R_f(x) \leq H(p \oslash q) \max\{p(x) - \check{p}(x), \, q(x) - \check{q}(x)\}. \tag{6}$$

## 3.3 Computing and Estimating Hoffman Constants

**The PVZ Algorithm.** In [17], the authors proposed a combinatorial algorithm to compute the precise value of the Hoffman constant for a matrix $A \in \mathbb{R}^{m \times n}$, which we refer to as the *Peña–Vera–Zuluaga (PVZ) algorithm* and sketch its main steps here.

**Definition 3.5.** A set-valued map $\Phi : \mathbb{R}^n \to \mathbb{R}^m$ assigns a set $\Phi(x) \subseteq \mathbb{R}^m$. The map is surjective if $\Phi(\mathbb{R}^n) = \cup_x \Phi(x) = \mathbb{R}^m$. Let $A \in \mathbb{R}^{m \times n}$. For any $J \subseteq \{1, 2, \ldots, m\}$, let $A_J$ be the submatrix of $A$ consisting of rows with indices in $J$. The set $J$ is called *$A$-surjective* if the set-valued map $\Phi(x) = A_J x + \{y \in \mathbb{R}^J : y \geq 0\}$ is surjective.

Notice that $A$-surjectivity is a generalization of linear independence of row vectors. We illustrate this observation in the following two examples.

**Example 3.6.** If $J$ is such that $A_J$ is full-rank, then $J$ is $A$-surjective, since for any $y \in \mathbb{R}^J$, there exists $x \in \mathbb{R}^n$ such that $y = A_J x$.

**Example 3.7.** Let $A = \mathbf{1}_{m \times n}$ be the $m \times n$ matrix whose entries are 1's. For any subset $J$ of $\{1, \ldots, m\}$ and for any $y \in \mathbb{R}^J$, let $x \in \mathbb{R}^n$ such that $\sum_i x_i \leq \min\{y_j, j \in J\}$. Then $y - A_J x \geq 0$. Thus any $J$ is $A$-surjective.

The PVZ algorithm is based on the following characterization of Hoffman constant.

**Proposition 3.8.** *[17, Proposition 2] Let $A \in \mathbb{R}^{m \times n}$. Equip $\mathbb{R}^m$ and $\mathbb{R}^n$ with norm $\|\cdot\|$ and denote its dual norm by $\|\cdot\|^*$. Let $\mathcal{S}(A)$ be the set of all $A$-surjective sets. Then*

$$H(A) = \max_{J \in \mathcal{S}(A)} H_J(A) \tag{7}$$

*where*

$$H_J(A) = \max_{y \in \mathbb{R}^m, \|y\| \leq 1} \min_{\substack{x \in \mathbb{R}^n \\ A_J x \leq y_J}} \|x\| = \frac{1}{\min_{v \in \mathbb{R}_+^J, \|v\|^* = 1} \|A_J^\top v\|^*}. \tag{8}$$

This characterization is particularly useful when $\mathbb{R}^m$ and $\mathbb{R}^n$ are equipped with the $\infty$-norm, since the computation of (8) reduces to a linear programming (LP) problem. The key problem is how to maximize over all $A$-surjective sets. To do this, the PVZ algorithm maintains three collections of sets $\mathcal{F}, \mathcal{I}$, and $\mathcal{J}$ where during every iteration: (i) $\mathcal{F}$ contains $J$ such that $J$ is $A$-surjective; (ii) $\mathcal{I}$ contains $J$ such that $J$ is not $A$-surjective; and (iii) $\mathcal{J}$ contains candidates $J$ whose $A$-surjectivity will be tested.

To detect whether a candidate $J \in \mathcal{J}$ is surjective, the PVZ algorithm requires solving

$$\min \|A_J^T v\|_1, \ \ s.t. \ v \in \mathbb{R}_+^J, \|v\|_1 = 1. \tag{9}$$

If the optimal value is positive, then $J$ is $A$-surjective, and $J$ is assigned to $\mathcal{F}$ and all subsets of $J$ are removed from $\mathcal{J}$. Otherwise, the optimal value is 0 and there is $v \in \mathbb{R}_+^J$ such that $A_J^\top v = 0$. Let $I(v) = \{i \in J : v_i > 0\}$ and assign $I(v)$ to $\mathcal{I}$. Let $\hat{J} \in \mathcal{J}$ be any set containing $I(v)$. Replace all such $\hat{J}$ by sets $\hat{J} \backslash \{i\}, i \in I(v)$ which are not contained in any sets in $\mathcal{F}$. The implementation used in our paper directly uses the MATLAB code provided by [17].

**Lower and Upper Bounds.** A limitation of the PVZ algorithm is that during each loop, every set in $\mathcal{J}$ needs to be tested, and each test requires solving a LP problem. Although solving one LP problem in practice is fast, a complete while loop calls the LP solver many times.

209 Here, we propose an algorithm to estimate lower and upper bounds for Hoffman constants. An
210 intuitive way to estimate the lower bound is to sample a number of random subsets from $\{1, \ldots, m\}$
211 and test for $A$-surjectivity. This method bypasses optimizing combinatorially over $\mathcal{S}(A)$ of $A$-
212 surjective sets and gives a lower bound of Hoffman constant by Proposition 3.8.

213 To get an upper of Hoffman constant, we use the result from [18].

214 **Theorem 3.9.** *[18, Theorem 4.2] Let $A \in \mathbb{R}^{m \times n}$. Let $\mathcal{D}(A)$ be a set of subsets of $J \subseteq \{1, \ldots, m\}$*
215 *such that $A_J$ is full rank. Let $\mathcal{D}^*(A)$ be the set of maximal elements in $\mathcal{D}(A)$. Then the Hoffman*
216 *constant measured under 2-norm is bounded by*

$$H(A) \leq \max_{J \in \mathcal{D}^*(A)} \frac{1}{\hat{\rho}(A_J)} \tag{10}$$

217 *where $\hat{\rho}(A)$ is the smallest singular value of $A$.*

218 Using the fact that $\| \cdot \|_1 \geq \| \cdot \|_2$, and the characterization from (8), we see that the upper bound also
219 holds when $\mathbb{R}^m$ and $\mathbb{R}^n$ are equipped with the $\infty$-norm. However, enumerating all maximal elements
220 in $\mathcal{D}(A)$ is not an improvement over enumerating $A$-surjective sets from a computational perspective.
221 Instead, we will retain the strategy as in lower bound estimation to sample a number of sets from
222 $\{1, 2, \ldots, m\}$ and approximate the upper bound by (10). We verify this approach via synthetic data.
223 The experiments are relegated to the Appendix.

# 4  Symmetry and the Fundamental Domain

225 In this section, we study a geometric characterization of the sampling domain for networks exhibiting
226 symmetry. This corresponds to *invariant neural networks*.

## 4.1  Linear Regions of Invariant Neural Networks

228 The notion of invariance for a neural network describes when a manipulation of the input domain
229 does not affect the output of the network. The manipulations we consider here are group actions.

230 **Definition 4.1.** Let $\sigma : \mathbb{R}^n \to \mathbb{R}$ be a piecewise linear function, and let $G$ be a group acting on the
231 domain $\mathbb{R}^n$. $\sigma$ is *invariant* under the group action of $G$ if for any element $g \in G$, $\sigma \circ g = \sigma$.

232 Given an invariant neural network, we can then define a sampling domain that takes into account the
233 effect of the group action.

234 **Definition 4.2.** Let $G$ be a group acting on $\mathbb{R}^n$. A subset $\Delta \subseteq \mathbb{R}^n$ is a *fundamental domain* if it
235 satisfies two following conditions: (i) $\mathbb{R}^n = \bigcup_{g \in G} g \cdot \Delta$; and (ii) $g \cdot \text{int}(\Delta) \cap h \cdot \text{int}(\Delta) = \emptyset$ for all
236 $g, h \in G, g \neq h$.

237 The fundamental domain of a group $G$ therefore provides a periodic tiling of $\mathbb{R}^n$ by acting on $\Delta$.
238 This is very useful in the context of numerical sampling for neural networks which are invariant
239 under some symmetry, since it means we can sample from a smaller subset of the input domain with
240 a guarantee to find all the linear regions in the limit. This allows us, in principle, to be able to use far
241 fewer samples while maintaining the same density of points.

242 **Theorem 4.3.** *Let $f : \mathbb{R}^N \to \mathbb{R}$ be a tropical rational map invariant under group $G$. Let $\Delta \subseteq \mathbb{R}^N$ be*
243 *a fundamental domain of $G$. Suppose $\mathcal{L}$ is the set of linear regions. Define the following two sets*

$$\mathcal{U}_c := \{A \in \mathcal{U} : A \subseteq \Delta\}$$
$$\mathcal{U}_n := \{A \in \mathcal{U} : A \cap \Delta \neq \emptyset\}.$$

244 *Then*

$$|G||\mathcal{U}_c| \leq |\mathcal{U}| \leq |G||\mathcal{U}_c| + \sum_{A \in \mathcal{U}_n \setminus \mathcal{U}_c} \frac{|G|}{|G_A|}.$$

245 *where $|G_A|$ is the size of the stabilizer of $A$.*

246 This gives us a method for estimating the total number of linear regions from sampling in the
247 fundamental domain using *multiplicity*, which we discuss next.

## 4.2 Sampling from the Fundamental Domain

To demonstrate the potential performance improvements in numerical sampling exploiting symmetry in the network architecture, we consider permutation invariant neural networks inspired by deep sets [19]. Our numerical sampling approach is inspired by very recent work in this area [20].

**Lemma 4.4** ([19]). *An $m \times m$ matrix $W$ acting as a linear operator of the form $W = \lambda I_{m \times m} + \gamma(\mathbf{1}^T \mathbf{1})$, where $\lambda, \gamma \in \mathbb{R}$ is permutation equivariant, meaning $WPx = PWx$ for any $x \in \mathbb{R}^m$, so it commutes with any permutation matrix.*

Using a weight matrix of this form, we can construct permutation invariant neural networks by setting the bias to 0, applying a ReLU activation after multiplication by $W$, and then summing. In this case, the network is invariant under the group action $S_n$, so the fundamental domain is the set of points with increasing coordinates, i.e., $\Delta = \{(x_1, \ldots, x_n) : x_1 \geq x_2 \geq \ldots \geq x_n\}$. This splits $\mathbb{R}^n$ into $n!$ tiles, so we have a clear and significant advantage in restricting sampling to the fundamental domain.

Note, however, that it is important to address the multiplicities of symmetric linear regions correctly: If a given Jacobian of shape $n \times 1$ has no repeated elements, this means it is contained in the interior of some group action applied to the fundamental domain. This means there are $n!$ total linear regions with this Jacobian. If, on the other hand, there are repeated coefficients in a given Jacobian $J$, we consider the set $C(J)$ of counts of repeated elements. For example, for $J = [1, 1, 0], C(J) = (2, 1)$. Then the multiplicity of a given Jacobian is given by

$$\text{mult}(J) = \frac{n!}{\prod_{c \in C(J)} c!}.$$

Using this multiplicity calculation we can efficiently estimate the number of linear regions while reducing the number of point samples by a factor of $n!$. This provides a dramatic gain in sampling efficiency.

In Figure 1, we present the results when Algorithm 2 is run with $R = 10, N = 10, M = 50$. These results show that the fundamental domain estimate performs well for low dimensional inputs but appears to overcount linear regions as $n$ increases. Despite divergence, there is still utility in this metric because we are often more concerned with obtaining an upper bound on the expressivity of a neural network than an exact figure and the fundamental domain estimate does not undercount the number of linear regions.

# 5 Symbolic Neural Networks

Here, we present the details on our practical contribution of a symbolic representation of neural networks as a new library integrated into OSCAR [15].

## 5.1 Computing Linear Regions of Tropical Puiseux Rational Maps

We present an algorithm that can compute the linear regions of *any* tropical Puiseux rational function. Intuitively, we do this by computing the linear regions of the numerator and denominator, and then considering intersections of such regions and how they fit together. Thus, a first step is to understand how the computation of linear regions works for tropical Puiseux polynomials. The key to our approach will be to exploit the polyhedral connection of tropical geometry and recast the problem in the language of polyhedral geometry. This, among other things, will allow us to make use of the extensive polyhedral geometry library in OSCAR [15] for implementation.

One important upshot from this study is that there is a strong connection between the number of linear regions of a tropical Puiseux rational function and the number of monomials that appear in its algebraic expression. Note, however, that the two are independent, in the sense that two Puiseux rational functions could have the same number of linear regions but different numbers of (nonzero) monomials, and conversely, the same number of monomials and a different number of linear regions. For instance, computing the number of linear regions requires some combinatorial data about the intersections of the polyhedra defined by monomials.

First, we need to know how to compute the linear regions of tropical polynomials. Let $P = \bigoplus_n a_n \odot x^n$ where by $x^n$ we mean $x_1^{n_1} \odot \cdots \odot x_k^{n_k}$ and powers are taken in the tropical sense. Then

as function $\mathbb{R}^k \to \mathbb{R}$, $P$ is given by $\max_n \{a_n + n_1x_1 \cdots + n_kx_k\}$. It follows that the linear regions of $P$ are precisely the sets of the form

$$S_n = \{x \in \mathbb{R}^n \mid a_m + m_1x_1 \cdots + m_kx_k \leq a_n + n_1x_1 \cdots + n_kx_k \text{ for all } m \neq n\}.$$

For any set $U$ on which $P$ is linear, we write $L(P, U)$ for the corresponding linear map. This gives us

$$L(P, S_n)(x) = a_n + n_1x_1 \cdots + n_kx_k. \tag{11}$$

We now rewrite (11) using polyhedral geometry. Recall that a polyhedron in $\mathbb{R}^k$ is a set of the form $P(A, b) = \{x \in \mathbb{R}^k \mid Ax \leq b\}$. We claim that each linear region is a polyhedron: For a fixed index $n$, define the matrix $A_n$ to be the $(N-1) \times k$ matrix whose rows are the vectors $m - n$, where $m$ ranges over the support of the coefficients of $P$ (ordered lexicographically) and $b_n$ to be the vector with entries $a_n - a_m$. Then $S_n = P(A_n, b_n)$. This gives us a way to encode the computation of the linear regions of tropical Puiseux polynomials using polyhedral geometry. As a direct consequence, intersections of linear regions of tropical Puiseux polynomials are also polyhedra. In particular, there are algorithms from polyhedral geometry for determining whether such polyhedra are realizable. One of the key observations given by our algorithm is that the linear regions of tropical Puiseux rational maps are *almost* given by $k$-dimensional intersections of the linear regions of the numerator and the denominator. Indeed, note that if $U$ is a linear region of $p$ and $V$ a linear region of $q$, then we have $L(U \cap V, p \oslash q) = L(U, p) - L(V, q)$. The only issue that arises is that there might be some repetition in the $L(U \cap V, p \oslash q)$ as $U$ ranges over the linear regions of $p$ and $V$ over the linear regions of $q$. In particular, linear regions of $p \oslash q$ might end up corresponding to unions of such $U \cap V$.

## 5.2 Computing Linear Regions

Determining the linear regions of a neural network may be approached *numerically* or *symbolically*. The numerical approach exploits the fact that linear regions of a neural network correspond to regions where the gradient is constant. Thus, to estimate the number of linear regions, we can evaluate the gradient on a sample of points (e.g., a mesh) in some large box $[-R, R]^n$. For sufficiently large $R$ and a sufficiently dense sample of points, we get an accurate estimate. The symbolic approach, on the other hand, exploits the connection between neural networks and tropical Puiseux rational maps. Indeed, we can symbolically compute a Puiseux rational map that represents the neural network and then compute the number of linear regions using the approach outlined in section 5.1.

To compare each method, we ran the computations on smaller networks with varying sizes to compare run times and precision. For the symbolic approach, we generate 20 neural networks with random weights for each architecture and then compute the tropical Puiseux rational function associated to each neural network and compute the linear regions using Algorithm 3.

For the numerical approach, we also work with synthetic data and generate 1000 neural networks with random weights for each architecture. We then estimate the number of linear regions in a box of size $[-10, 10]^n$ and sample 1000 points from this domain.

In both cases, we use He initialization for the weights, i.e., we generate weights with distribution $N(0, \frac{2}{\sqrt{d}})$ where $d$ is the input dimension. The data we obtain in this manner is summarized in Tables 10 and 11. For the symbolic approach, we also track the number of nonzero monomials to compare this quantity with the number of linear regions. For networks with 3 layers, we find the numerical estimate to be quite close, but for 4 it seems to diverge. This could be because in the numerical approach, we are only counting the number of unique Jacobians that can be found in the domain. A situation could arise where the same linear function is disconnected and hence counted twice by the symbolic approach but only once for the numerical approach.

The main observations from our experimental study are as follows. The numerical approach is faster, but offers no guarantee of precision: When running the computation for a given $R$ and mesh grid, there seems to be no easy way of determining whether we have indeed hit all the linear regions or whether we have obtained an accurate estimate of the arrangements of these regions. It is possible to either overestimate or underestimate the number of linear regions. In particular, there is a priori no obvious way to select the parameters. We found the symbolic approach to be more precise, but slower. In general, the number of monomials seems to be far larger than the number of linear regions, which contradicts the intuition of Figure 2.

Both algorithms suffer from the curse of dimensionality: in the case of the numerical approach, the number of samples in a meshgrid grows exponentially with respect to the dimension. In the case of

the symbolic approach, calculations with polytopes seem to scale poorly with dimension and with the complexity of the neural network.

## 6   Discussion: Limitations & Directions for Future Research

In this paper, we set up a framework to interpret and analyzed the expressivity of neural networks using techniques from polyhedral and tropical geometry. We demonstrated several ways in which a symbolic interpretation can often enable computational optimizations for otherwise intractable tasks and provided new insights into the inner workings of these networks. To the best of our knowledge, ours is the first work to provide practical tropical geometric theory and algorithms to numerically compute and analyze the expressivity of a neural network both in terms of inherent neural network quantities as well as tropical geometric quantities.

Despite the theoretical and practical advancement of tropical deep learning that our work offers, it is nevertheless subject to limitations, which we now discuss and which inspire directions for future research.

**Experimental Limitations.**   The curse of dimensionality is a common theme in deep learning, and our work is unfortunately no exception. The methods introduced in this paper are quite fast for small enough networks, but scale poorly with dimension and more complex architectures.

We note that the main computational bottlenecks of the Puiseux rational function associated with a neural network are the implementation of fast multivariate Puiseux series operations. Our current computations rely on a custom implementation of this type of operation, and one potential avenue for improvement would be using such methods once they have been implemented in OSCAR [15].

For the computation of linear regions, both the numerical and symbolic approaches suffer from the curse of dimensionality. For instance, the numerical approach requires sampling on a mesh grid in a box of the form $[-R, R]^n$ where $n$ is the input dimension. In particular, the number of points needed is proportional to the volume, which scales exponenially in $n$. Similarly, the symbolic approach relies on the computation of the Puiseux rational function associated with a neural network and polytope computations, both of which are challenging computational problems in higher dimensions.

Most of our computations rely on carrying out some elementary computations many times. Thus, another avenue of improvement for this would be to parallelize.

**Structural Limitations.**   Much of what we are studying are basically framed as a combinatorial optimization problem, which are known to be difficult. In particular, computing the Hoffman constant is equivalent to the Stewart–Todd condition measure of a matrix and both quantities are NP-hard to compute in general cases [17, 21].

Further studying and understanding where and how symbolic computation algorithms can be made more efficient, e.g., by parallelization, would make our proposed approaches more applicable to larger neural networks. Our work effectively proposes a new intersection of symbolic computation and deep learning, so there remains infrastructure to set up to make methods from these two fields compatible.

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

## A  Further Experimental Details

We ran the final computations on NVIDIA GeForce RTX 3090 GPUs. Table 7 lists the time taken by
each experiment. Given that our experiments do not include training on large datasets, the experiments
are not particularly expensive from the perspective memory usage, and all the code can be run on a
laptop. The detail provided in the paper correspond roughly to the amount of computational resources
that were used for this work, omitting trial and testing runs.

## B  Algorithms

---
**Algorithm 1** Lower and approximate upper bound of Hoffman constant

---
**Require:** $A$: an $m \times n$ matrix; $B$ max number of iterations; $\epsilon$ threshold of testing surjectivity.
 1: Initialize $H_L = H^U = 0$.
 2: **for** $i \in 1, \ldots, B$ **do**
 3:      Sample a random integer $K$.
 4:      Sample a random subset $J$ from $\{1, \ldots, m\}$ of size $K$.
 5:      Solve (9). Let $t$ be the optimal value;
 6:      **if** $t > \epsilon$ **then**
 7:          $J$ is surjective. Update $H_L = \max\{H_L, \frac{1}{t}\}$;
 8:      Compute the minimal singular value of $\hat{\rho}(A_J)$;
 9:      **if** $\hat{\rho}(A_J) > 0$ **then**
 10:         Update $H^U = \max\{H^U, \frac{1}{\hat{\rho}(A_J)}\}$;
      **return** Lower bound $H_L$ and approximate upper bound $H^U$.

---

---
**Algorithm 2** Estimation of the ratio of fundamental domain sampling to regular sampling

---
**Require:** The input dimension $n$, $R \in \mathbb{R}$ side length for cube centered at the origin from which the
     samples are taken, $M$ number of models to use, $N$ base number of points to sample.
 1: **for** $m \in 1..M$ **do**
 2:      Create a permutation invariant model $\sigma$ with input dimension $n$.
 3:      Sample $N^n$ points in the cube with side length $R$ centered at the origin. Note that the number
     of points in the sample grows exponentially with the input dimension $n$.
 4:      Compute the Jacobian matrices of the network at each point, round to 10 decimal place to
     avoid numerical errors, remove duplicates, and count the number of unique Jacobians.
 5:      Sample $\frac{N^n}{n!}$ points from the fundamental domain of $\mathbb{R}^n$ intersected with the sampling cube.
 6:      Compute the unique Jacobians similarly as for the regular sampling.
 7:      Sum the multiplicities of each Jacobian to get an estimate of the total number of linear
     regions.
 8:      Record the ratio of the fundamental domain estimate to the regular estimate.
      **return** The average ratio across $M$ models.

---

## C  Proofs

### C.1  Proof of Proposition 3.4

*Proof.* The polyhedra defined by

$$\left( \begin{bmatrix} A_p \\ A_q \end{bmatrix} - \mathbf{1} \begin{bmatrix} a_{i_p} \\ a_{j_q} \end{bmatrix} \right) x \le \begin{bmatrix} b_{i_p}\mathbf{1} - b_p \\ b_{j_q}\mathbf{1} - b_q \end{bmatrix}$$

form a convex refinement of linear regions of $f$. Let

$$\text{res}_{i_p, j_q}(x) := \left( \begin{bmatrix} A_p \\ A_q \end{bmatrix} - \mathbf{1} \begin{bmatrix} a_{i_p} \\ a_{j_q} \end{bmatrix} \right) x - \begin{bmatrix} b_{i_p}\mathbf{1} - b_p \\ b_{j_q}\mathbf{1} - b_q \end{bmatrix}$$

denote the residual of $x$ to the polyhedron. We have

$$R_f(x) \le H(p \oslash q) \max\{\|\text{res}_{i_p, j_q}(x)_+\|_\infty : 1 \le i_p \le m_p \,; 1 \le j_q \le m_q\}.$$

---

**Algorithm 3** Linear regions of tropical Puiseux rational functions

---

**Require:** Tropical Puiseux polynomials $p, q$ in $n$ variables.
1: Compute the linear regions $U_1, \ldots, U_l$ of $p$, and set $L_i = L(p, U_i)$.
2: Compute the linear regions $V_1, \ldots, V_m$ of $q$, and set $S_j = L(q, V_j)$.
3: Compute the pairs $(i, j)$ such that $U_i \cap V_j$ has dimension $n$
4: **for** $(i, j)$ such that $U_i \cap V_j$ has dimension $n$ **do**
5:     Compute the linear map $T_{ij} = L_i - S_j$

6: Set $S$ to be the set of all $T_{ij}$
7: **for** $T \in S$ **do**
8:     Compute the set $I(T)$ indices $(i, j)$ such that $T = T_{ij}$.
9:     Compute the set $C(T)$ of connected components of

$$\bigcup_{(i,j) \in I(T)} U_i \cap V_j$$

    **return** $\bigcup_{T \in S} C(T)$.

---

---

**Algorithm 4** Numerical estimation of neural network linear regions

---

**Require:** The architecture of a linear activation neural network $\sigma$ with scalar output, $R \in \mathbb{R}$ side length for cube centered at the origin from which the samples are taken, $M$ number of models to use, $N$ number of points to sample.
1: **for** $m \in 1..M$ **do**
2:     Create a model with architecture $\sigma$ and initialise weights and biases using He inialisation.
3:     Sample $N$ points in the cube with side length $R$ centered at the origin.
4:     Compute the Jacobian matrices of the network at each point.
5:     Round the Jacobians matrices to 5 decimal places to avoid floating point errors.
6:     Remove duplicates and count the number of unique Jacobians.
    **return** The average number of linear regions.

---

443   Note that

$$\|\text{res}_{i_p, j_q}(x)_+\|_\infty = \left\| \left( \begin{bmatrix} A_p x + b_p - \mathbf{1}(a_{i_p} x + b_{i_p}) \\ A_q x + b_q - \mathbf{1}(a_{j_q} x + b_{j_q}) \end{bmatrix} \right)_+ \right\|_\infty$$

$$= \max_{k, \ell} \left\{ (A_p x + b_p)_k - (a_{i_p} x + b_{i_p}), \ (A_q x + b_q)_\ell - (a_{j_q} x + b_{j_q}), \ 0 \right\}$$

$$= \max \left\{ p(x) - (a_{i_p} x + b_{i_p}), \ q(x) - (a_{j_q} x + b_{j_q}), \ 0 \right\}$$

444   Therefore,

$$\max_{i_p, j_q} \|\text{res}_{i_p, j_q}(x)\|_\infty = \max_{i_p, j_q} \left\{ p(x) - (a_{i_p} x + b_{i_p}), \ q(x) - (a_{j_q} x + b_{j_q}), \ 0 \right\}$$

$$= \max \left\{ p(x) - \min_{i_p} \{a_{i_p} x + b_{i_p}\}, \ q(x) - \min_{j_q} \{a_{j_q} x + b_{j_q}\}, \ 0 \right\}$$

$$= \max \left\{ p(x) - \check{p}(x), \ q(x) - \check{q}(x) \right\}$$

445   which proves (6).     □

446   **C.2   Proof of Lemma 3.3**

447   *Proof.* From the definition of minimal effective radius we have

$$R_f(x) = \min\{r : B(x, r) \cap U_i \neq \emptyset, U_i \in \mathcal{U}\} = \min\{r : d(x, U_i) \leq r, U_i \in \mathcal{U}\}$$
$$= \max\{d(x, U_i) : U_i \in \mathcal{U}\}.$$

448   For each linear region $U_i$ characterized by $\widetilde{A}_{U_i} x \leq \widetilde{b}_{U_i}$, by (1), $d(x, U_i) \leq H(\widetilde{A}_{U_i}) \|(\widetilde{A}_{U_i} x - \widetilde{b}_{U_i})_+\|$.
449   Passing to maximum we have

$$R_f(x) = \max_{U_i \in \mathcal{U}} d(x, U_i) \leq \max_{U_i \in \mathcal{U}} H(\widetilde{A}_{U_i}) \max_{U_i \in \mathcal{U}} \|(\widetilde{A}_{U_i} x - \widetilde{b}_{U_i})_+\| = H(f) \max_{U_i \in \mathcal{U}} \|(\widetilde{A}_{U_i} x - \widetilde{b}_{U_i})_+\|.$$

450     □

| Lower bounds $H_L$ | 0.5460 | 0.1520 | 0.6220 | 0.5771 | 0.1208 | 0.0844 | 1.0389 | 0.1492 |
|---|---|---|---|---|---|---|---|---|
| Time $t_L$ | 0.2495 | 0.2449 | 0.2446 | 0.2458 | 0.2443 | 0.2463 | 0.2466 | 0.2477 |
| True values $H$ | 0.3298 | 0.7980 | 0.3772 | 0.8376 | 6.5934 | 2.6744 | 0.9372 | 1.2645 |
| Time $t$ | 0.2132 | 0.1504 | 0.1253 | 0.1722 | 0.1566 | 0.1529 | 0.1721 | 0.1568 |
| Upper bounds $H^U$ | 0.2081 | 0.5090 | 0.2903 | 1.0539 | 3.8508 | 1.3942 | 0.5484 | 0.8031 |
| Time $t^U$ | 0.0043 | 0.0040 | 0.0040 | 0.0040 | 0.0040 | 0.0040 | 0.0040 | 0.0040 |

Table 1: Hoffman constants, lower bounds, approximate upper bounds, and their corresponding computational time for $m_p = 10$, $m_q = 5$ and $n = 3$

## C.3 Proof of Theorem 4.3

*Proof.* The action of $G$ partitions $\mathcal{U}$ into a set of orbits $[\mathcal{U}]$ and we have $|\mathcal{U}| = \sum_{[A] \in [\mathcal{U}]} |[A]|$.

From property (i) defining a fundamental domain $\cup_{A \in \mathcal{U}} A = \cup_{\sigma \in G} \sigma \cdot \Delta$, we have the trivial lower bound on the number of linear regions $|\mathcal{U}| \geq \sum_{A \in \mathcal{U}_c} |[A]| = |G||\mathcal{U}_c|$. Let $A \in \mathcal{U}$. By Lagrange's theorem, the orbit of $A$ is such that $|[A]||G_A| = |G|$. Thus we have

$$|\mathcal{U}| \leq \sum_{A \in \mathcal{U}_n} |[A]| \leq |G||\mathcal{U}_c| + \sum_{A \in \mathcal{U}_n \setminus \mathcal{U}_c} \frac{|G|}{|G_A|}.$$

$\square$

# D   Numerical Calculations of the Hoffman Constant

We illustrate the computation of Hoffman constant of tropical Puiseux rational map on synthetic data. We generate tropical Puiseux rational maps by randomly generating two tropical Puiseux polynomials $p$ and $q$. Specifically, suppose $p$ has $m_p$ monomials and $q$ has $m_q$ monomials. We construct an $m_p \times n$ matrix $A_p$ and an $m_q \times n$ matrix $A_q$ by uniformly sampling entries from $[0, 1]$. We then form the matrix defined by (4). We then compute the exact Hoffman constant using the PVZ algorithm and estimate its lower bound and approximate its upper bound by our proposed algorithm. We record the computation time and the number of calls to solve the LP problem in the whole loop.

In the experiment we take different values of $m_p$, $m_q$, $n$ and $B$. For each of the parameters we repeat all computation for 8 times. The true Hoffman constants, lower bounds, upper bounds, and the computation time per linear region can be found in Table 1,2,3, and the number of iterations of the PVZ algorithm and average time to solve LP during each iteration can be found in Table 4,5,6. From the tables we can see that computing the true Hoffman constants requires testing surjectivity and solving over thousands LP problems, which costs a lot of time. Although the lower bounds and approximate upper bounds can be loose, the computational time is much faster for lower bounds and upper bounds, which implies its potential to apply for real data applications.

# E   Tables

Tables 8 and 9 summarise the outcomes of the experiments on the computation of linear regions for tropical Puiseux rational functions. For a fixed number of variables $n_{\text{var}}$ and number of monomials $n_{\text{monomials}}$, we generate $n_{\text{samples}}$ random Puiseux rational functions by picking random coefficients and exponents using Julia's inbuilt random number generation functions, where both the numerator and the denominator have $n_{\text{monomials}}$ monomials. We then compute the number of linear regions for each of these rational functions and take the average over our all the samples that were generated.

# F   Figures

| Lower bounds $H_L$ | 2.4965 | 5.9002 | 2.3501 | 3.7049 | 1.1434 | 0.8335 | 1.6517 | 2.2396 |
|---|---|---|---|---|---|---|---|---|
| Time $t_L$ | 0.8924 | 0.9101 | 0.9127 | 0.8914 | 0.9132 | 0.9154 | 1.1117 | 0.6190 |
| True values $H$ | 26.2231 | 726.8115 | 173.0057 | 23.8868 | 52.6080 | 8.1573 | 8.5050 | 18.7593 |
| Time $t$ | 6.0048 | 2.3452 | 5.5451 | 3.6778 | 3.2828 | 2.9109 | 3.5494 | 1.7530 |
| Upper bounds $H^U$ | 8.2854 | 323.5149 | 21.3290 | 7.4338 | 183.8179 | 254.1373 | 36.7961 | 32.5276 |
| Time $t^U$ | 0.0136 | 0.0137 | 0.0143 | 0.0132 | 0.0120 | 0.0139 | 0.0148 | 0.0097 |

Table 2: Hoffman constants, lower bounds, approximate upper bounds, and their corresponding computational time for $m_p = 15$, $m_q = 9$ and $n = 6$

| Lower bounds $H_L$ | 0.1120 | 0.1382 | 0.1227 | 63.9169 | 0.2331 | 0.1191 | 0.0571 | 0.1126 |
|---|---|---|---|---|---|---|---|---|
| Time $t_L$ | 0.2622 | 0.2628 | 0.2625 | 0.2672 | 0.2771 | 0.2715 | 0.2689 | 0.2633 |
| True values $H$ | 0.0017 | 1.2683 | 1.5375 | 0.0832 | 0.2777 | 0.3537 | 0.0464 | 0.1586 |
| Time $t$ | 0.0112 | 0.0217 | 0.1002 | 0.0182 | 0.0582 | 0.0693 | 0.0189 | 0.0122 |
| Upper bounds $H^U$ | 10.6826 | 1.7551 | 3.2794 | 7.2134 | 26.4648 | 2.6868 | 25.0251 | 5.1308 |
| Time $t^U$ | 0.0775 | 0.0789 | 0.0780 | 0.0784 | 0.0789 | 0.0782 | 0.0782 | 0.0769 |

Table 3: Hoffman constants, lower bounds, approximate upper bounds, and their corresponding computational time for $m_p = 15$, $m_q = 5$ and $n = 7$

| # iterations | 94 | 86 | 67 | 83 | 99 | 86 | 75 | 83 |
|---|---|---|---|---|---|---|---|---|
| Time per LP | 0.0042 | 0.0026 | 0.0025 | 0.0026 | 0.0025 | 0.0025 | 0.0026 | 0.0026 |

Table 4: Number of iterations in the PVZ algorithm and average time to solve LP during each iteration for $m_p = 10$, $m_q = 5$ and $n = 3$

| # iterations | 2437 | 1110 | 1731 | 1441 | 1432 | 1706 | 1741 | 1095 |
|---|---|---|---|---|---|---|---|---|
| Time per LP | 0.0152 | 0.0093 | 0.0092 | 0.0098 | 0.0098 | 0.0102 | 0.0095 | 0.0097 |

Table 5: Number of iterations in the PVZ algorithm and average time to solve LP during each iteration for $m_p = 15$, $m_q = 9$ and $n = 6$

| # iterations | 2 | 607 | 525 | 80 | 194 | 355 | 78 | 19 |
|---|---|---|---|---|---|---|---|---|
| Time per LP | 0.0027 | 0.0027 | 0.0026 | 0.0027 | 0.0032 | 0.0027 | 0.0028 | 0.0027 |

Table 6: Number of iterations in the PVZ algorithm and average time to solve LP during each iteration for $m_p = 15$, $m_q = 5$ and $n = 7$

| Experiment | Compute time |
|---|---|
| Linear regions of tropical Puiseux rational functions (3 variables) | 4.7 hours |
| Linear regions of tropical Puiseux rational functions (4 variables) | 13.3 hours |
| Symbolic linear regions computation for neural networks | 35 minutes |
| Numeric linear regions computation for neural networks | 4.9 minutes |
| Sampling on fundamental domain | 13 minutes |

Table 7: Compute details

| $n_{\text{monomials}}$ | Average number of regions | Average runtime |
|---|---|---|
| 20 | 84.4 | 4.0 seconds |
| 50 | 160.1 | 11.6 seconds |
| 100 | 264.2 | 29.5 seconds |
| 200 | 375.2 | 66.0 seconds |
| 350 | 500.8 | 139.2 seconds |
| 500 | 580.8 | 202.7 seconds |
| 800 | 706.1 | 394.8 seconds |
| 1000 | 776.2 | 563.6 seconds |

Table 8: Computation for $n_{\text{var}} = 3$, $n_{\text{samples}} = 12$

| $n_{\text{monomials}}$ | Average number of regions | Average runtime |
|---|---|---|
| 20 | 157.5 | 12.6 seconds |
| 50 | 398.75 | 50.3 seconds |
| 100 | 667.75 | 83.8 seconds |
| 200 | 1021.5 | 237.5 seconds |
| 350 | 1614.5 | 987.3 seconds |
| 500 | 1909.5 | 1682.1 seconds |
| 800 | 2432.0 | 3436.7 seconds |
| 1000 | 2876.5 | 5441.8 seconds |

Table 9: Computation for $n_{\text{var}} = 4$, $n_{\text{samples}} = 4$

| Architecture | Average number of linear regions | Average number of monomials | Average runtime(s) |
|---|---|---|---|
| $[2, 2, 1]$ | 3.85 | 5.75 | 0.4166 |
| $[4, 3, 1]$ | 6.75 | 9 | 0.4646 |
| $[4, 4, 1]$ | 14.2 | 13.55 | 1.5794 |
| $[3, 2, 2, 1]$ | 6.8 | 30.15 | 1.7679 |
| $[3, 3, 2, 1]$ | 17.55 | 176.75 | 97.9659 |

Table 10: Symbolic computation

| Architecture | Average number of linear regions | Average runtime(s) |
|---|---|---|
| $[2, 2, 1]$ | 3.041 | 0.01667 |
| $[4, 3, 1]$ | 6.339 | 0.01667 |
| $[4, 4, 1]$ | 11.936 | 0.01667 |
| $[3, 2, 2, 1]$ | 3.549 | 0.01683 |
| $[3, 3, 2, 1]$ | 7.381 | 0.01678 |

Table 11: Numerical computation

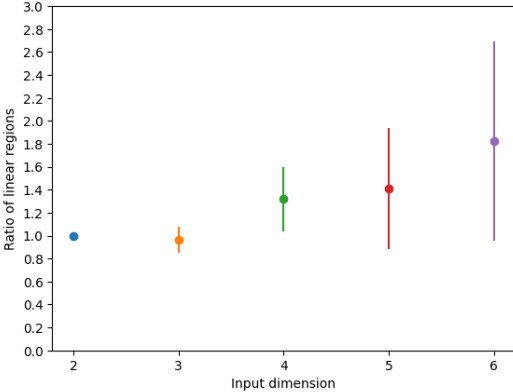

Figure 1: Ratio estimates for different input sizes with standard deviation error bars

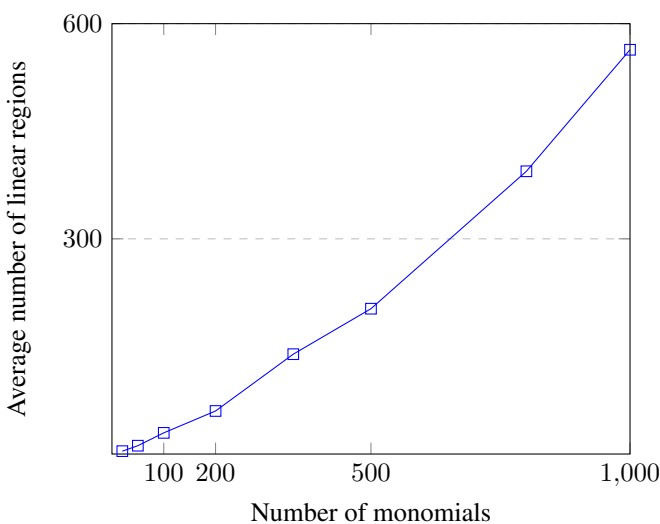

Figure 2: Linear regions of a Puiseux rational function in 3 variables

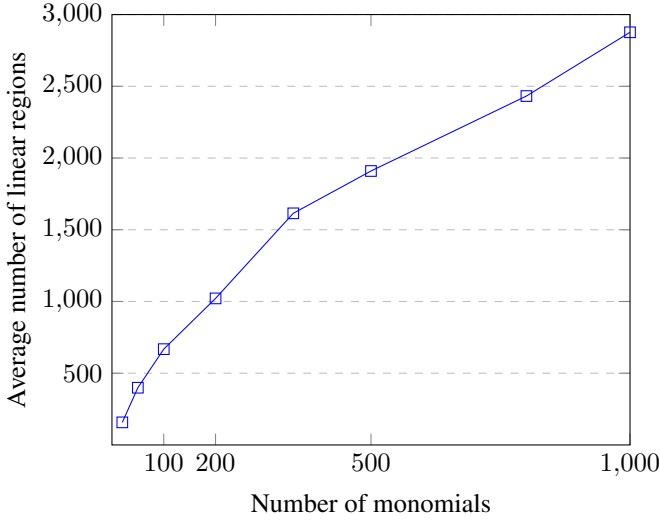

Figure 3: Linear regions of a Puiseux rational function in 4 variables

