# OpenReview forum: "Tropical Expressivity of Neural Networks"
_NeurIPS.cc/2024/Conference — Submitted to NeurIPS 2024_

### Official Review · Reviewer_hdMP · 2024-07-10

**Soundness:** 1
**Presentation:** 2
**Contribution:** 1
**Rating:** 3
**Confidence:** 5

**Summary:**

This paper proposes new methods to count the number of linear regions in neural networks by viewing them as tropical Puiseux rational maps. By computing their Hoffman constant, the authors are able to identify a sampling radius which ensures that all the network’s linear regions will be intersected. They use this insight to propose algorithms for counting the number of linear regions for both invariant and traditional networks.

**Strengths:**

The paper is well-written with virtually no typos and errors. The technical content is accessible and not unnecessarily convoluted and the proofs and concepts are presented clearly.

**Weaknesses:**

The main weakness of the work, in my opinion, can be summarized in the following points:

- the connections to tropical geometry and group theory are not rigorous beyond the point of simple notational fixes
- the motivation for the work and how it fills gaps in the existing literature is unclear, and
- the effective utility of the approach is not convincingly demonstrated by the theory or experiments.

**Rigor of tropical and group theories**

I believe this point is the biggest weakness of the paper. From the perspective of tropical algebra, vectors and polynomials live in $\bar{\mathbb{R}}$. The authors mention $\bar{\mathbb{R}}$ in line 87, but then this is never used in most of their work. This might seem as a notational fix, but it is not, as it introduces problems in virtually every single result in the paper. This first becomes a real issue in (4), where maximums are taken over, potentially, $\infty$. How is it guaranteed that (4) exists in the context of tropical algebra? This is a recurring problem that appears in (5), (6), and (7). Another important issue at the intersection of group theory and tropical algebra is how groups are defined. Semirings, by construction, are objects that do not admit additive inverses. This means that, if one wants to define groups on such structures, great care needs to be taken as to how groups are defined, how they act on vector spaces, and what groups are actually permissible in this context.

From the perspective of group theory how is the group action defined? How do the group elements act on vectors in tropical spaces? Group representations $\rho: G \to \operatorname{GL}$ require the concept of an invertible matrix, however that concept is ill-defined in tropical vector spaces.

(Moreover, the authors define incorrectly $\bar{\mathbb{R}}$. The infinity element needs to be the identity element of tropical addition: if one opts to use the max-plus semiring, $-\infty$ should be used. If we use the min-plus semiring, $\infty$ should be used. However, this is a notational fix.)

**Motivation**

In terms of motivation it is unclear how the work is related to the existing works. There have been countless results on the number of linear regions of neural networks, and quite a few results using tropical geometry at that. What void in the literature does this paper fill? The related work paragraph lists some of the works in tropical geometry, but doesn’t highlight where these works come short and how the proposed manuscript fills that void. Moreover, there is no discussion of why the existing works on linear counting that do not utilize tropical geometry are also not able to handle the presented context.

**Effective utility**

At the end of the day, I’m not sure I understand what the utility of the method is. Ignoring here the questions on motivation, the goal is to make deep learning more interpretable. However, the authors’ experiments diverge when input sizes are larger than $6$ and the networks are deeper than $4$ layers. Modern deep learning uses input sizes significantly larger than $100$ and decade old networks are deeper than $10$ layers. So what effectively are we learning about deep networks?

Unfortunately, there are some larger issues with the method and experiments. Beyond the concerns from the perspective of tropical geometry, we have zero guarantees about the upper and lower bounds of the Hoffman constant. There is no asymptotic analysis on the sample complexities of the bounds, no analyses about tightness or optimality, or even, at the very least, an analysis that the bounds are not vacuous or trivial. On line 272 the authors claim that even though their estimate diverges, that’s acceptable because frequently we’re interested is an upper bound on expressivity. However, how can we guarantee that there the number of regions is not undercounted (I couldn’t find a proof)? For that statement to be true the bound needs to be tight, but from their own experiments (Tables 1 and 2), the true $H$ ends up being larger than the upper bound, which is used to calculate the radius and eventually the radius. Obviously, then, the computed bounds are not representative: then, since the estimate Hoffman constant is not accurate and the algorithm diverges, what essentially do we gain?

**Questions:**

I have some more fine-grained comments and questions.

**Major**

There is a recurring discussion about massively improving efficiency. However, that efficiency is when compared against what? There is not a single comparison against existing works, so the improvement is over what? Regardless, as the authors discuss in Algorithm 2 and in the section about limitations, there is an *exponential* scaling with the number of inputs. Even if the authors introduce a factorial reduction for invariant networks, they require exponential sampling, which leads to no real improvement, as is evident by their really small parameter values and large running times. How where these sample numbers chosen in the first place? I couldn’t find any asymptotic analysis that guarantees with high probability that the whole space will be sampled with such a number of samples. Overall, the computational complexity of the approach is prohibitive, as it required $N^n$ Jacobian computations, which is nigh impossible for parameter values found in real networks.

There is a significant lack of attribution to prior work. In the introduction, textbooks on tropical geometry are cited as well as papers on the counting of linear regions. However, when the intersection of tropical geometry and machine/deep learning is discussed, zero references are provided, whereas the field has been fairly active in the last 5 years with dozens of highly cited papers. Similarly, in the related works section the final citation has no connection to the presented paper beyond being at the intersection of tropical geometry and machine learning. However, there are much older and much more well-cited papers to include as general references on the field, both as general reviews and as specific applications (tropical compression of neural networks comes to mind). As a final comment on this train of thought, would the authors like to elaborate on line 352? Specifically, both [11] and [12] analyze the expressivity of neural networks using tropical geometric quantities (through zonotopes and Newton polytopes). How do the authors view their work as the first in doing that?

As a last major comment, there is another lack of preciseness, however to a lesser extend as to what I mentioned above. There is a recurring discussion about one method being more precise than the other: however, what does that mean? Preciseness is never defined, and it’s not straightforward how the authors compute the ground truth of the number of linear regions. For example, how can we say that Table 10 is more precise than Table 11? In a similar vain, there are underlying assumptions that are not discussed or communicated. For example, what are the assumptions of 4.3 from the perspective of group theory? Does the theorem work for any, possibly infinite group? Similarly, there are constraints in Definition 3.5 that are not communicated. The way A-surjectivity is introduced, it is required that $m < n$. However, that makes explicit assumptions about the networks that can be analyzed, which is never communicated or considered.

**Minor**

There are some minor things that are unclear in the text. For example, it is unclear what Section 5.2 refers to. Is it a table or a plot, and which one? In Theorem 4.3 $\mathcal{L}$ is mentioned which is never used or defined, and it is unclear what the term bias is referring to in 256 (I’m assuming $\lambda$?). Table 1 (and others) are lacking headers and it’s impossible to understand what the different columns represent. Finally, the term maximal elements is used with respect to sets of sets. However, sets lack canonical order, so it’s unclear what “maximal element” means in this context.

Overall the notation is good, but at a few points it is confusing. Matrix indices usually refer to columns, not rows, which can lead to confusion and the choice of $\rho$ for the singular values (over the universally accepted $\sigma$) can hurt readability. In 329 the input dimension is denoted by $d$, in contrast to the rest of the manuscript and the notation $\min\{y_j, j\in J\}$ is not consistent with set theory nor optimization (and also in conflict with the authors own notation, for example, in (10) or 295. Finally, $S_n$ is used as the permutation group but also as a set in 295.

**Limitations:**

I think the authors accurately identify the main limitations of their approach, which relates to the large computational complexity.

---

> ### Author Rebuttal · Authors · 2024-08-07
>
> We thank the reviewer for their thorough reading of our work and detailed feedback. We would like to address weaknesses raised by the reviewer.
> ### Rigor of Tropical Geometry and Group Theory
> We apologize for the confusion caused: while we focus on the tropical geometric interpretation of neural networks, we are actually not working exclusively in the tropical setting, because some of the concepts would not make sense, as pointed out by the reviewer.
> Our results do not require us to work in $\overline{ℝ}$. We never work with tropical
> vector spaces, only tropical polynomials, which can be understood without
> reference to $\overline{ℝ}$. In (4), we are taking a maximum over a finite set of Hoffman
> constants, each of which lies in $ℝ$ rather than $\overline{ℝ}$ (one concern here might be
> that we are allowing coefficients of the tropical Puiseux rational function to be $−∞$, but we can ignore these). Similar observations can be made about (5), (6), and (7).
> On the matter of group actions, we never claim to be working with group actions in a tropical sense. In our submission, we are not working with tropical vector spaces and group actions, but simply with $ℝ^n$ and group actions in the usual sense.
> ### Motivation
> We have conducted a second literature review and found one citation missing from our original submission: *On the Decision Boundaries of Neural Networks: A Tropical Geometry Perspective* by Alfarra et al. (2022). We apologize for this oversight. This reference provides important insights that demonstrate the value of tropical geometry in deep learning. To the best of our understanding our work does not overlap with theirs in either theoretical focus or novel computational contributions.
>
> As far as we know, all existing computational methods for finding the number of linear regions of a neural network rely on sampling which doesn’t guarantee an exact solution, or impose some restrictions such as boundedness on the input variables rather than the total number of linear regions.
> 1. We provide new tools for computing the size of the domain needed to count the linear regions, and we give a new method for computing the exact number of linear regions of a neural network on an unbounded domain.
> 2. We provide a direct way of computing a novel representation of neural networks, namely, the tropical Puiseux rational form. This opens the door to further connections with tropical geometry such as the number of (non-redundant) monomials that appear in the tropical expression of a neural network.
> ### Effective Utility
> The main goal of our paper was to present a proof of concept, not to develop highly optimized code. We believe there is room for improvement (particularly for symbolic computations), and better algorithms could greatly boost our approach’s performance, which we noted in the Discussion section as a future research direction. We focused on smaller networks for simpler analysis. However our code can handle more complex architectures than those we used. Our symbolic method can process networks with inputs of dimension 784 in a minute (Table 1 in the attachment).
>
> Finally, the reviewer noticed that the two methods we outlined yield different results. The discrepancy is because the numerical algorithm may undercount the number of linear regions in certain cases due to not taking into account connectivity of regions. We will clarify this in the revision.
>
> We thank the reviewer for identifying errors in Table 1–3 regarding the computed Hoffman constants. After reviewing the public code by Peña et al. (2018), we found that it was incorrect. We have since computed the values by brute force, and the updated tables show that the lower bounds are always below the true values. Regarding upper bounds, Theorem 3.9 states that the Hoffman constant is bounded by the inverse of the smallest nonzero singular value of submatrices of $A$. We used a threshold of $10^{−10}$ to determine nonzero values, but found that many small singular values ranged between $10^{−30} ∼ 10^{−14}$. This confirms that the singular values provide upper bounds. We agree that these bounds can be tightened as a direction for future research. This study could also be promising for new results concerning the Stewart–Todd condition measure of a matrix.
>
> ### Improving Efficiency
> Our claim of improving efficiency refers to the factorial reduction in scaling for invariant networks. We believe that the ideas we introduce in this manuscript may also lead to similar improvements for other methods, in a symmetric setting.
> ### Lack of Attribution to Prior Work
> We do not claim that this work is the first to analyze expressivity of neural networks using tropical geometry; we explicitly mention previous work in this direction in our manuscript in lines 39–41. We are not aware of work that implements algorithms to compute the *exact* number of linear regions of a network with unbounded inputs.
> ### Lack of Precision
> To clarify, the symbolic method for computing linear regions yields the ground truth, by converting the network to a tropical Puiseux rational function and applying Algorithm 3.
>
> For missing assumptions, we acknowledge that Theorem 4.3 is missing the assumption that the group is finite. We believed it was implicit in the context (cardinal arithmetic would be implausible in the context of machine learning). Moreover, we disagree that our definition of $A$-surjective sets introduces any new assumptions on $m$ and $n$. The definition makes sense for any matrix $A$ and the assertion that the existence of $A$-surjective sets implies that $m<n$ is incorrect, as such sets exist for all $A≠0$.
>
> We thank the reviewer for flagging typos and inaccuracies; these will be addressed in the revision.
>
> **References:**
> Alfarra, M., Bibi, A., Hammoud, H., Gaafar, M., & Ghanem, B. (2022).
> *On the decision boundaries of neural networks: A tropical geometry perspective.* IEEE Transactions on Pattern Analysis and Machine Intelligence, 21 45(4), 5027–5037

---

> > ### Comment · Reviewer_hdMP · 2024-08-11
> >
> > Thank you for the rebuttal and for engaging with the reviewing process.
> >
> > Regarding the tropical geometry, I have to say I'm confused by the response. If the Puiseux polynomials are interpreted using $\mathbb{R}$, and all the other concepts in the paper are independent of tropical geometry, then what is the connection to tropical geometry?

---

> ### Author Response · Authors · 2024-08-11
>
> We thank the reviewer for reading our rebuttal and giving us this important opportunity to clarify our approach further.
>
> ### The role of infinity
>
> The tropical semiring is a fundamental algebraic object in tropical geometry where all other constructions are based. For the sake of completeness and correctness, we must introduce $-\infty$ to the extended real line since it is the neutral element for tropical addition. In our application to machine learning, the main way the tropical framework appears is that it gives a very convenient (and, as our work demonstrates, computable) way of representing neural networks. In other words, we are interested in *tropical* representations these functions. While tropical Puiseux polynomials naturally yield functions defined over the whole extended real line, this is generally not the case for neural networks, and thus we usually restrict to $\mathbb{R}$. We would also like to emphasize that while the mathematics are mostly developed over the real numbers, a large portion of our code uses OSCAR's implementation of the tropical algebra.
>
> ### Connection to tropical geometry
>
> Tropical geometry is a broad field and intersects with many other mathematical fields including polyhedral geometry, algebraic geometry, combinatorics, etc. For example, tropical Puiseux polynomials, tropical monomials are naturally related to linear regions and polyhedra in ways which are fully elaborated in our paper. The connection between neural networks and tropical geometry is *crucial* since it allows us to use OSCAR to develop symbolic computations.
> While we acknowledge that we don’t use any advanced tropical geometry for our results (and instead use techniques from other fields e.g., polyhedral geometry), we maintain that our work is nevertheless tropical geometric in nature: Firstly, our results are built on the tropical geometric interpretation of neural networks. Secondly, we principally interpret tropical Puiseux polynomials as functions over $\mathbb{R}$, which is the natural approach for studying their function-theoretic properties as we do in our work. Despite this interpretation, they still are rigorously tropical objects.
>
> We understand that the mix of tropical and classical approaches and perspectives in our work may have inadvertently introduced confusion, which we will carefully clarify in the revision.
>
> ### Responding to other concerns
> We are very grateful for this opportunity to engage with the reviewer who gave us a very detailed, thoughtful, and constructive review, which we appreciated very much. We would like to ask if the reviewer has any other questions pertaining to our rebuttal of the concerns in the original report? We are aware that our rebuttal was brief, given that we were constrained to the quite strict character limit, but we would very much like to engage further and respond in greater detail to the reviewer’s observations in the original report above.

---

> ### Author Response · Authors · 2024-08-12
>
> As we approach the end of the discussion period, we would like to take this opportunity to address some additional concerns raised in the reviewer’s original report. Due to the character limit for rebuttals, our initial response had to be concise, and we regret not being able to provide the level of detail that the depth of the original report warrants.
>
> It is important to clarify that some statements made in the review above are incorrect. Additionally, as with reviewer vnGc, we are concerned that our significant contributions of adapting symbolic computation to the study of neural networks were not fully recognized, given the lack of acknowledgment and comments.
>
> ### Rigor of Tropical Geometry and Group Theory
>
> We apologize for the confusion raised in our work, since, indeed, while we focus on the tropical geometric interpretation of neural networks, we are actually not working fully and exclusively in the tropical setting throughout our work, precisely because some of the concepts would not make sense, as pointed out by the reviewer.  We now elaborate on the specific concerns raised by the reviewer.
>
> Our results do not require us to work over $\overline{\mathbb{R}}$. In particular, the absence of $\overline{\mathbb{R}}$ is intentional: we never work with tropical vectors or vector spaces, but only with tropical (Puiseux) polynomials, which can be understood with no reference to $\overline{\mathbb{R}}$.  Referring specifically to the expressions mentioned by the reviewer: in (4), we are taking a maximum over a finite set of Hoffman constants, each of which lies in $\mathbb{R}$ rather than $\overline{\mathbb{R}}$ (one concern here might be that we are in principle allowing coefficients of the tropical Puiseux rational function to be $-\infty$, but we can simply ignore such coefficients). Similar observations can be made about (5), (6), and (7), which should now clarify that these expressions are well-defined.
>
> We respond similarly to the concern about group actions in the tropical setting: we never claim to be working with group actions in a tropical sense. In our submission, we are not working with tropical vector spaces and group actions, but simply with $\mathbb{R}^n$ and group actions in the classical sense.
>
> In our revision, we will explicitly state that these specific concepts should not be evaluated tropically so as to diffuse any misunderstandings.
>
> ### Motivation
>
> We thank the reviewer for this question and would like to take this opportunity to respond to concerns raised by the reviewer.
>
> With regard to motivation and contribution, we respond by stating that to the best of our knowledge, all computational methods (that have been implemented!) for computing the number of linear regions of a neural network resort to some form of sampling that doesn't guarantee an exact solution, or impose some restrictions such as boundedness on the input variables (for instance, as our Jacobian sampling method does) rather than computing the *total* number of linear regions. Our work fills this void in two different ways:
> 1. We provide new tools for computing the size of the domain that needs to be restricted to in order to count the correct number of linear regions, and we give a new method (the so-called symbolic method) for computing the exact number of linear regions of a neural network on an unbounded domain;
> 2. We also provide a new avenue for understanding the expressivity of neural networks (or more generally, studying interpretability) - specifically, we provide a direct way of computing another representation of neural networks, namely, the tropical Puiseux rational form. This representation is important because it opens the door to using other theory from tropical geometry, e.g., computing other measures of complexity, such as the number of (non-redundant) monomials that appear in the tropical expression of a neural network.
>
>
>
> (continued in the next official comment)

---

> ### Author Response · Authors · 2024-08-12
>
> ### Effective Utility
>
> Firstly, we want to clarify that the main goal of our paper was to present a proof of concept, not to develop highly optimized code for large-scale stress tests on modern deep neural networks. Our focus was on laying the theoretical groundwork and demonstrating the feasibility of our approach. Therefore, we did not prioritize optimizing certain parts of the code for performance. This was a deliberate choice to highlight the direct applicability of symbolic computation techniques to neural networks. For example, in the tropical Puiseux polynomial library, we implemented simple algorithms, rather than using more sophisticated computer algebra methods. We believe there is considerable room for improvement in this area, and more efficient algorithms could significantly boost our approach's performance, which we noted in the Discussion section as a future research direction.
>
> Secondly, our experiments mainly focused on smaller networks to allow for clear and understandable analysis. We believe that understanding small networks is helpful for making deep learning more interpretable. However, our code is capable of handling more complex architectures than those used in our experiments. For instance, our symbolic method can process neural networks with inputs of dimension of 28 × 28 (the dimension of inputs in the MNIST dataset) in about 1 minute. This shows that our approach is scalable, contrary to what our smaller experiments might suggest. We have added new tables in the attachment to demonstrate this on various architectures with higher-dimensional inputs (see Table 1). In our revision, we will include these larger-scale results alongside the smaller-scale experiments, as we believe the smaller-scale ones provide a clearer and more accessible proof of concept, allowing us to illustrate the key insights and mechanisms of our approach more easily.
>
> Finally, the reviewer raised an important issue regarding the two methods we outlined for computing linear regions, which yield different results for networks deeper than 4 layers. We appreciate the reviewer for pointing this out and giving us the chance to clarify. The discrepancy is because the numerical algorithm version implemented in our submission allows for disconnected linear regions, while the symbolic method accounts for connectivity. In other words, the numerical algorithm may undercount the number of linear regions for certain networks. We will address this discrepancy in the revision.
>
> We thank the reviewer for their careful reading and for identifying errors in Table 1--3 regarding the computed Hoffman constants. The lower bounds should indeed be below the true Hoffman constants. After reviewing the public code by Peña et al. (2018), we found that it returned incorrect values, leading to the errors. We have since re-computed the values using a brute force method, and the updated tables show that the lower bounds are always below the true values, consistent with the theory.
>
> Regarding upper bounds, Theorem 3.9 (line 214 of our original submission) states that the Hoffman constant is bounded by the inverse of the smallest nonzero singular value of submatrices of $A$. We used a threshold of $10^{-10}$ to determine nonzero values, but found that many small singular values ranged between $10^{-30} \sim 10^{-14}$. This confirms that the singular values provide upper bounds, aligning with the theory. We agree with the reviewer that these bounds can be tightened, and we will mention this as a direction for future research in our revision. This study could also be promising for new results concerning the Stewart--Todd condition measure of a matrix.
>
> We now turn to addressing the questions asked by the reviewer.  We summarize the major concerns in subtopics.
>
> ### Improving efficiency
>
> We would like to clarify that our claim of improving efficiency refers to the factorial reduction for invariant networks, when compared to naively sampling without taking the symmetry into account. In particular, we do not claim to introduce computational improvements on already existing works, although we believe that the ideas we introduce in this manuscript may also lead to similar improvements for other methods.
>
> How our methods scale and the question of their utility in the context of modern deep learning have been addressed earlier in this response.
>
> (continued in the next official comment)

---

> > ### Author Response · Authors · 2024-08-12
> >
> > ### Lack of attribution to prior work
> >
> > We are confused by the reviewer's claim that ''when the intersection of tropical geometry and machine/deep learning is discussed, zero references were provided'' when these appear in a subsection on related work (lines 39-48 of our submission).
> >
> > To clarify our intentions on line 352, we are certainly not claiming that this work is the first to analyze expressivity of neural networks using tropical geometric quantities; we explicitly reference previous work in this direction in our manuscript in lines 39-41 (''Tropical geometry has been used to characterize deep neural networks with piece wise linear activation functions, including two of the most popular and widely-used activation functions, namely, rectified linear units (ReLUs) and maxout units.''). However, we are not aware of previous work that provides and implements exact algorithms that compute the total number of linear regions of a network-this is what the sentence on line 352 is stating.
> >
> > ### Lack of precision
> >
> > We now address the reviewer's comments about the lack of precision in our comparison between our two methods: it is useful to state explicitly that the symbolic method for computing linear regions does already compute the ground truth. As a reminder, this is based on Algorithm 3, which computes the *exact* number of linear regions of a tropical Puiseux rational function. More specifically, the symbolic method converts the neural networks to a tropical Puiseux rational function, and then applies Algorithm 3.
> > With that in mind, our claim that Table 10 in our original submission is more precise than Table 11 is communicating the fact that **the numerical method (used for Table 11) does not always give the exact answer (because, for instance, some regions might be missed), while the symbolic method does.**
> >
> > As for missing assumptions, we acknowledge that Theorem 4.3 is missing the assumption that the group is finite.
> >  We left this assumption out because it is implicit in the context (for instance, the statement is clearly implausible in the context of machine learning if it is to be interpreted in terms of cardinal arithmetic). Moreover, we disagree that our definition of $A$-surjective sets introduces any new assumptions on $m$ and $n$.  **The definition makes sense for any matrix $A$ (with no assumption that $m < n$) and the claim that the existence of $A$-surjective sets implies that $m < n$ is incorrect: For any non-zero matrix $A$, there exists an $A$-surjective set of row indices. In particular, our use of this notion does not introduce any additional assumptions on the networks we consider.**

---

> > > ### Comment · Reviewer_hdMP · 2024-08-12
> > >
> > > Thank you for the discussion. Regarding the incorrect statements in the review, you seem to be referring to a single sentence (of a large paragraph, of an even larger section, of the **Questions** subfield). The whole purpose of the **Questions** subfield is to ask clarifying questions, which I did. I was operating under the assumption that for any choice of $J$ the set should be non-trivial, which you clarified in your rebuttal. There is little value in attempting to discredit a thorough review for asking a clarifying question.
> > >
> > > At the same time, there are many points raised in my review that are still not addressed, from theoretical ones (how are group actions and group representations defined in this work), to practical ones (what are the different columns in the tables). You mention you work with group actions in the classical sense (which, itself, is not descriptive), but there is no mention of group representations and how these actions act on the objects of vector spaces.
> > >
> > >
> > > ### Tropical geometry
> > > It is still unclear why tropical geometry is *necessary* for the paper. OSCAR is a computational algebra tool that does not require tropical geometry. Puiseux polynomials can exist outside of tropical geometry. All the computational tools do not require any concepts from tropical geometry, but rather, as the authors suggest, polyhedral geometry. This extends beyond using the extended reals: in (Zhang, 2018) and (Charisopoulos, 2018) the Newton polytopes are studied, which are central objects in tropical geometry. It appears that this work could avoid using tropical notation, and the paper would remain almost identical.
> > >
> > > ### Effective utility
> > > No one is expecting production level code, but asking for a method that can work on networks of reasonable (in the order of 20) depths and reasonable (in the order of 1000) input sizes does not impose such requirements. Even in the new experiments you provided, the network sizes are tiny and there is a large variance in the running time (you mention 1 minute, however some instances run for over 10 minutes for a 3 layer network with a total of 4 hidden and one output neuron). We are in agreement that understanding small networks can lead to more profound understanding of deep learning, but I strongly disagree a 5 neuron network is a large-scale experiment: in fact, I would argue such a network doesn't even satisfy the bar of a small network. As I mentioned in my original review, there is no asymptotic analysis nor a statement about the tightness of the bounds. In essence, running the proposed method on a network may or may not reveal something about the network, and we have no way of knowing how accurate the method might be. The majority of the paper is devoted to the numerical method (Section 3), it is expected that comments will be made on what most of the work is about.
> > >
> > > ### Lack of attribution
> > > The comment in the review was precise: "when the intersection of tropical geometry and machine/deep learning is discussed, zero references were provided". The intersection is introduced in 33 and no general references on the field are provided. 39-48 are works related to this one, and I already mentioned relevant papers which are more related to the work than, for example, [13] and [14].

---

> ### Author Response · Authors · 2024-08-13
>
> We appreciate the time and effort the reviewer has invested in evaluating our work. However, we were concerned by the tone of the recent exchange, which we believe deviated from the professional and respectful dialogue we value. Our intent in pointing out mathematical inaccuracies was not to discredit the review but to ensure that our work is assessed with the utmost accuracy. We hope to steer the conversation back to a constructive and respectful engagement and remain grateful for the opportunity to further clarify and discuss important aspects of our research.
>
> We also want to clarify that our response was not limited to a single sentence/paragraph in **Questions** section. We noted that there were additional incorrect statements and misunderstandings other than the ones we highlighted in our previous response. Our intention is to address these comprehensively, ensuring that our work is accurately represented and understood. We appreciate your attention to detail and the opportunity to clarify these points.
>
> ### Group actions
> Our original reading of the section in the review about group actions was that there was some confusion about how group actions are defined in the tropical setting, to which we answered that our group actions are not tropical. Group actions are quite a basic concept, but for clarification, we think it is worth being explicit about what we mean. By a left group action of $G$ on a set $X$, we mean a map $(g, x) \mapsto g \cdot x$ that satisfies the following properties:
> 1. For any $x$ in $X$ and $g, h$ in $G$, we have $(g h) \cdot x = g \cdot (h \cdot x)$
> 2. For all $x$ in $X$, we have $1_G \cdot x = x$.
>
> A right group action can be defined in a similar way (see, e.g., the Wikipedia page on group actions). In accordance with standard mathematical terminology, we use the term \emph{group action} to refer to a map $G \times X \to X$ that is either a left or a right action. In particular, we are making \emph{no} assumptions on the group $G$ beyond finiteness. The group $G$ acting on $\mathbb{R}^n$ further induces an action on the finite set of linear regions $\mathcal{U}$. In this case, for a linear region $x\in \mathcal{U}$ and a group element $g\in G$, the action of $g$ sends $x$ to another linear region $g\cdot x$. Thus we are in the case of finite groups acting on finite sets, without more complicated group representation theory involved. In particular, we do not mention group representations since they are not relevant here: in this level of generality, there is no natural way attaching a representation $\rho : G \to \mathrm{GL}(\mathbb{R}^n)$ to the group action $G \circlearrowright \mathbb{R}^n$ as the reviewer seems to be suggesting. It is of course possible to define certain group actions using representations, which gives rise to the class of *linear* group actions, but in our work there is no need to impose this additional restriction since it is not necessary for our results to hold. Given that this is seemingly a source of confusion, we will clarify this in the revision.
>
> ### Tables
> We apologise for the lack of clarity in Tables 1-6. As explained in Appendix D, we compute the Hoffman constant together with upper and lower bounds for 8 different Puiseux rational functions, and each column of Tables 1-3 corresponds to the outputs given for of these 8 samples (which are randomly generated with different structural parameters for each table). Tables 4-6 provide more detail about compute times, and again each column corresponds to a different sample. We will clarify this in the revision.
>
> ### Tropical geometry
>
> The reviewer is philosophically correct in observing that the entire paper could theoretically be rewritten without reference to tropical geometry, however, in the same line of reasoning, *any* mathematical theory could be rephrased starting from set theory. Doing so would diminish the significance of the specific mathematical contributions that tropical geometry brings to our work. Our choice to use the language of tropical geometry was twofold: (i) it provides a unifying and systematic framework for the general study of expressivity of neural networks; and (ii) it builds upon and extends the work of Zhang et al. (2018), where the main contribution of that paper was to provide an alternative perspective to studying neural networks through the lens of tropical geometry.  Furthermore, we would like to claim that the use of tropical geometry is in agreement with other well established works in the linear regions literature, and does not obscure the mathematics of our paper.
>
> Finally, concerning the OSCAR implementation, we agree that in principle our code could be reimplemented without using tropical notions. However, one of the four cornerstones of the OSCAR library is precisely the tropical geometry library (building from polymake) and our work opens the door for researchers to apply these very powerful comptuational tools to study neural networks.

---

> > ### Author Response · Authors · 2024-08-13
> >
> > ### Effective utility
> > 1. We don't claim the new experiments are large scale, simply that they are larger scale than the original ones.
> > 2. Concerning the scale, we would also like to point out that that other papers in the area work with small, shallow networks, and have large compute times; see e.g., https://arxiv.org/pdf/1810.03370 and https://arxiv.org/pdf/1711.02114.
> > 3. We are confused about the reviewer's comment concerning our symbolic method: ''As I mentioned in my original review, there is no asymptotic analysis nor a statement about the tightness of the bounds. In essence, running the proposed method on a network may or may not reveal something about the network, and we have no way of knowing how accurate the method might be.''
> > We acknowledge that such considerations are sensible for our numerical method, but these concerns are not relevant to the symbolic method, where, by the very nature of symbolic computation, **our method *does* reveal something about the network, and is *fully accurate*.**
> > 4. While Section 3 does present one of the important contributions of our work, it is certainly not the only contribution we make and while we do appreciate feedback on this portion of the work that occupies 3 out of the 9 allowed content pages, we would like to point out that the entirety of Section 5 is also an important contribution because it presents our symbolic method and how it compares to the numerical one.  As mentioned above, there are fundamental differences to the two approaches and while there exist other numerical approaches in the literature, **ours is the first to propose symbolic approaches which essentially mitigate all of the concerns raised by the reviewer** on interpretability and accuracy (which are indeed important questions in numerical computation).
> >
> > ### Lack of attribution
> > Thank you for your feedback. We would like to respectfully address the comment regarding the absence of references when discussing the intersection of tropical geometry and machine/deep learning. **The statement that ''zero references were provided'' is entirely false**, particularly considering the subsequent clarification that ''no general references on the field are provided.'' We have indeed cited relevant works within this intersection, including on topics beyond expressivity and linear region counting.
> >
> > There are other references (listed below) around the intersection of tropical geometry and machine learning, but most of these do not directly overlap with the focus of our work, so we are not convinced that including them would be any more helpful than some of the existing references (which the reviewer already raised concerns on in terms of relevance to our work). Two notable ones that we do agree to include are Maragos, Charisopoulos, and Theodosis (2021), which provides a detailed survey of the general field, and Montufar, Ren, and Zhang (2022), which uses tropical geometry to study the linear regions of neural networks with neural networks with maxout units. We would like to point out that among the list below, the paper that we agree to include by Brandenburg et al. (2024) falls under the category of ''contemporaneous work'' by NeurIPS (as it appeared online within 2 months of our submission). If there are other particular papers that the reviewer believes we have overlooked that are directly relevant to our work, we would be grateful for the opportunity to review and incorporate them.
> >
> > **Additional references:**
> >
> > Maragos, Charisopoulos, and Theodosis. *Tropical Geometry and Machine Learning*, 2021.
> >
> > Brandenburg, Loho, and Montúfar. *The Real Tropical Geometry of Neural Networks*, 2024.
> >
> > Smyrnis and Maragos. *Tropical Polynomial Division and Neural Networks*, 2019.
> >
> > Montufar, Ren, and Zhang. *Sharp bounds for the number of regions of maxout networks and vertices of minkowski sums*, 2022.

---

> > ### Comment · Reviewer_hdMP · 2024-08-13
> >
> > Hello again,
> >
> > ### Group theory
> > I'm familiar with abstract group theory, group theory on topological spaces, representation and character theory. I appreciate the clarification. As $\mathbb{R}^n$ is a topological space (where standard definitions define a group acting on such a space as $g \cdot f(x) = f(g^{-1}x)$, see Dehmamy et al., 2021 or Cohen et al., 2019), a vector space (where standard definitions include group representations and generally $g \cdot f(x) = \rho(g) f(g^{-1}x)$, see Cohen and Welling, 2017 or Weiler and Cesa, 2019), and a set (where abstract group definitions could potentially suffice), stating that you define group actions in the classical sense is not descriptive, hence why I asked for the clarification. It's still not clear to me how you practically define the group action, but at this point I think at least the theoretical concepts are covered.
> >
> > ### Tropical geometry
> > I understand the authors perspective, but I disagree. Zhang et al., 2018 was explicitly using concepts from tropical geometry, which is not present in this work.
> >
> > ### Effective utility
> > 1. These considerations were for the numerical method as they explicitly mentioned the Hoffman constant and the point still stands.
> > 2. The technical content of the paper spans pages 3 to 8. Three of those 5 pages are devoted to the numerical method, one page to Section 4, and one page to Section 5. The majority of the technical content is on the Hoffman constant.
> >
> > ### Lack of attribution
> > The statement is not false. Please read the full statement, printed here for your convenience: "In the introduction, textbooks on tropical geometry are cited as well as papers on the counting of linear regions. However, when the intersection of tropical geometry and machine/deep learning is discussed, zero references are provided, whereas the field has been fairly active in the last 5 years with dozens of highly cited papers". The comment explicitly mentions the introduction and where the intersection of tropical geometry and machine learning is first introduced. To further avoid confusion, I explicitly stated that general textbooks and cited papers are given for each of the fields separately, as a way to explicitly specify what I was referring to. As a final guard to misinterpretation, the following sentence reads "Similarly, in the related works section the final citation has..." which further reinforces that the previous part was not talking about the related works section. I further explained this in my previous response, and the statement is correct, I do not understand why this is such a point of contention. I think the listed works in your response act as adequate general references.
> >
> > Dehmamy et al., 2021: Automatic Symmetry Discovery with Lie Algebra Convolutional Network
> >
> > Cohen et al., 2019: A General Theory of Equivariant CNNs on Homogeneous Spaces
> >
> > Cohen and Welling, 2017: Steerable CNNs
> >
> > Weiler and Cesa, 2019: General E(2) - Equivariant Steerable CNNs

---

> > > ### Author Response · Authors · 2024-08-13
> > >
> > > We would once again like to thank the reviewer for taking the time to reply promptly to our previous responses.
> > >
> > > ### Group actions
> > > In light of the reviewer's response, we believe there has been a misunderstanding in our convention, and we would like to clarify the following points:
> > > 1. In section 4.1, we work with *arbitrary* group actions, in  the sense that we are assuming we have been given a group $G$ together with an action map $G \times \mathbb{R}^n \to \mathbb{R}^n$ that satisfies the properties we mentioned in our previous response.
> > > 2. In section 4.2, we work with *permutation invariant neural networks*, i.e., networks that are invariant under permutations of their inputs. In other words, we are working with a specific (and standard!) group action of the symmetric group $S_n$ on $\mathbb{R}^n$ called the *natural permutation representation*, given by $\sigma \cdot (x_1, \dotsc, x_n) = (x_{\sigma(1)}, \dotsc, x_{\sigma(n)})$ (i.e., the $i$-th coordinate of $x$ becomes the $\sigma(i)$-th coordinate of $\sigma \cdot x$). This is a linear representation and can thus also be defined in terms of a representation of $S_n$.
> > >
> > > We would also like to point out that the formulas the reviewer provided in their response appear to be definitions of *examples* of actions rather than definitions of the *notion* of an action, which is what we are working with. For example, the first formula given by the reviewer appears to be taken from equation (1) in Dehmamy et al. (2021); this defines an *specific* action on the space of functions $f : \mathcal{S} \to \mathbb{R}^n$ where $\mathcal{S}$ is some topological space. Notice that the formula given is also already assuming we are given a group action on the space $\mathcal{S}$.
> > >
> > > ### Tropical geometry
> > > We acknowledge that the second main contribution by Zhang et al. (2018) is the use of theory from tropical geometry to derive an analytic upper bound on the number of linear regions.  The first is a representation of neural networks in terms of tropical rational functions, which, strictly speaking, is defined by tropical algebra (in the sense that tropical algebra is used to define these functions).  We are following suit: while our interpretation of neural networks are tropical Puiseux rational maps is tropical algebraic, we maintain that our framework is *tropical*.  We would like to add that tropical algebra was first introduced and studied in the context of tropical geometry, rather than as an independent theory (even though since the introduction of tropical geometry, work on tropical linear algebra, for instance, has been well-studied).  See Maclagan and Sturmfels, *An Introduction to Tropical Geometry* (2015) and Joswig, *Essentials of tropical combinatorics*' (2021).
> > >
> > > ### Effective utility
> > > We appreciate the reviewer's observations regarding the lengths of the sections in our paper. While it is true that the symbolic section is shorter than the part on the numerical estimate, it nonetheless constitutes a significant portion of the main technical content and is highlighted as one of the key contributions of our paper in the introduction as well as the conclusion. Therefore, we feel it would be more balanced to give it due consideration in the review. For context, we would also like to note that the reviewer provided detailed feedback on Section 4, which is shorter than Section 5, indicating that length alone should not diminish the importance of the content.

---

> > > > ### Author Response · Authors · 2024-08-13
> > > >
> > > > ### Lack of attribution
> > > > Thank you for your detailed feedback. We were happy to see that we have found common ground concerning which additional references to include in our revision. We appreciate the opportunity to clarify our position regarding the inclusion and placement of references related to the intersection of tropical geometry and machine/deep learning.
> > > >
> > > > We agree with your observation that the section titled Tropical Geometry does not contain specific references to works applying tropical geometry to study machine learning, except in the context of expressivity. This is because the focus of that section is to *introduce* tropical geometry and its relevance to deep learning.
> > > >
> > > > The references to works that explore the intersection of tropical geometry and machine learning more broadly are indeed included in the Related Work section, which follows immediately after. We intended the Related Work section to provide a comprehensive view of the current literature, including those intersections. While we understand that this might be interpreted differently, we hope this clarifies our intent.
> > > >
> > > > We recognize that both the Tropical Geometry and Related Work sections touch on the intersection of tropical geometry and machine learning. However, we believe our initial structure effectively distinguishes between the introduction of the concept and the detailed review of the relevant literature. Thus, the statement that "when the intersection of tropical geometry and machine/deep learning is discussed, zero references are provided" is imprecise, as this could easily be interpreted as applying to both paragraphs taken together. Both sections indeed discuss this intersection, with more references provided in the latter.
> > > >
> > > > Finally, we feel it is important to address the characterization of this issue as a "major concern'' (it is mentioned in the strongly-worded statement as a "significant lack of attribution to prior work''). While we acknowledge the importance of proper citation placement, **we believe this is a minor structural issue rather than a substantive flaw in our work**, since the citations *did* appear in our work, placed subsequently in the text, though, we concede, in different sections. We are fully committed to improving the clarity and structure of our paper and we are happy to include the agreed-upon additional references, but we also want to emphasize that this issue does not, in our view, diminish the overall contribution or rigor of our work.  This is the reason why, to us, it is "such a point of contention.''

---

### Official Review · Reviewer_R5Gh · 2024-07-12

**Soundness:** 3
**Presentation:** 2
**Contribution:** 2
**Rating:** 6
**Confidence:** 3

**Summary:**

This study investigates the expressive power of deep fully-connected ReLU networks (or a piecewise linear function) from the perspective of tropical geometry. The number of linear regions gives an estimate of the information capacity of the network, and the authors provide a novel tropical geometric approach to selecting sampling domains among linear regions.

**Strengths:**

- An effective sampling domain is proposed as a ball of radius bounded by Hoffman constant, a combinatorial invariant

- The proposed sampling algorithm is doable and implemented.

**Weaknesses:**

- The proposed algorithms suffer from the curse of dimensionality

**Questions:**

- The authors mention in the abstract that the number of linear regions is an estimate of information capacity of the network. I need more clarifications, because this fact bridges tropical geometry and machine learning study.

- (minor) l.133: I was a bit confused here. Is the matrix-vector product $Ax$ in the sense of tropical algebra or in the ordinal sense? What does a “vector-inequality” $Ax \le b$ mean?

**Limitations:**

it is discussed in Section 6

---

> ### Author Rebuttal · Authors · 2024-08-06
>
> We would like to thank the reviewer for their feedback on our work. We are
> pleased that the reviewer found our work to be sound and would like to take
> this opportunity to respond to the concerns raised.
>
> ### Linear Regions as an Estimate of Information Capacity
>
> In choosing the number of linear regions in the domain of a neural network as our measure of its expressivity, we are following the precedent set by seminal existing work, such as those by Zhang et al. (2018), Montúfar et al. (2014), and Raghu et al. (2017). To briefly summarize the motivation, the idea is that by counting these linear regions, we get an estimate of how many classes a neural network could classify in theory if the weights were ‘hand-tuned,’ so to speak. It is also a measure of the model’s flexibility—the more linear pieces there are, the easier it is to approximate smooth functions, in the spirit of Riemann integration, for example.
>
> ### Matrix Product Notation
>
> We apologize for any confusion caused by our choice of notation on matrix products. In Section 3.1 on line 133 of our submission, the matrix product refers to classical matrix multiplication as opposed to tropical matrix multiplication. In general, throughout our submission, we did not work with tropical linear algebra. The inequality is used component-wise, so that we are considering the set of points such that each component satisfies the inequality. In practice, this means we are taking the intersection of a set of linear inequalities in the components of x, which is useful for defining a polyhedron.
>
> ### Scalability and the Curse of Dimensionality
>
> We now turn to addressing to the weakness of the curse of dimensionality raised by the reviewer. In our work we presented a novel algorithm for using symmetry in the domain to improve sampling efficiency to numerically estimate the linear regions. Indeed, we are aware that our technique suffers from the curse of dimensionality, which we explicitly discussed in the Limitations section. Despite the computational complexity being exponential in the number of inputs, we believe that our contribution still has academic value for two reasons: Firstly, it lays the ground for further investigation in using symmetry to accelerate difficult computations; the arguments concerning the fundamental domain are very general and can be applied to any symmetric neural network. Secondly, while the overall computational complexity may not be improved, the case we investigate in our submission does lead to a factorial improvement in complexity, as it scales with the input dimension, which is a nontrivial improvement.
>
> For our algorithm which computes the symbolic representation of a neural network and uses it to compute the exact number of linear regions, the scaling is much better, allowing up to 784-dimensional inputs. We point out that this achieves the dimension of important and widely-used benchmarking inputs, such as MNIST. The technique already scales well enough to yield interesting results concerning the number of monomials and linear regions, and with further algorithmic improvements and better hardware utilization (which are directions for future research that we identified in our original submission), it would likely improve even further.
>
> Thank you again for your feedback on our work and the opportunity to respond to questions and weaknesses. We hope we were able to clarify the reviewer’s concerns.
>
> **References:** (cited in our original submission)
>
> Liwen Zhang, Gregory Naitzat, and Lek-Heng Lim. *Tropical geometry
> of deep neural networks.* In International Conference on Machine Learning
> (2018), pages 5824–5832. PMLR (2018).
>
> Guido F. Montúfar, Razvan Pascanu, Kyunghyun Cho, and Yoshua Bengio. *On the number of linear regions of deep neural networks.* In Advances
> in Neural Information Processing Systems (2014), 27 (2014).
>
> Maithra Raghu, Ben Poole, Jon Kleinberg, Surya Ganguli, and Jascha
> Sohl-Dickstein. *On the expressive power of deep neural networks.* In International Conference on Machine Learning (2018), pages 2847–2854. PMLR
> (2017).

---

> ### Comment · Reviewer_R5Gh · 2024-08-12
>
> Thank you for detailed responses. I will keep my score as is.
>
> > Linear Regions as an Estimate of Information Capacity
> > ... by counting these linear regions, we get an estimate of how many classes a neural network could classify in theory if the weights were ‘hand-tuned,’ ...
>
> Thank you for clarifying it. I skimmed Zhang et al. (2018), Montúfar et al. (2014), and Raghu et al. (2017), but none of the authors called it information. Is the number of linear regions equivalent to either Shannon's mutual information (aka channel capacity), the information bottleneck in the rate-distortion theory, or any other information measure? If not, I recommend the authors not to call it 'information' to avoid miscommunication.

---

> > ### Author Response · Authors · 2024-08-12
> >
> > Thank you for your continued engagement with our work and for the thoughtful feedback.
> >
> > To the best of our knowledge, the number of linear regions in a neural network is not equivalent to any of the information measures listed by the reviewer. We understand the potential for miscommunication and will be more careful with our terminology in the future to ensure clarity and precision.
> >
> > Thank you once again for your valuable input.

---

### Official Review · Reviewer_vnGc · 2024-07-15

**Soundness:** 3
**Presentation:** 3
**Contribution:** 3
**Rating:** 3
**Confidence:** 3

**Summary:**

The paper studies the expressivity of neural networks as captured by the number of linear regions using tools from tropical geometry. There are three main contributions, two of which are theoretical and the other is about open source library that allows the analysis of neural networks as Puiseux rational maps.

The first theoretical contribution is that they propose a new approach for selecting sampling domains among the linear regions and the second is a way to prune the search space for network architectures with symmetries.

Prior work on tropical geometry and deep neural nets have analyzed ReLU and maxout units. Contrary to prior works, this work makes an effort to understand the geometry of the linear regions not just their number. To do so the authors propose a way of sampling the domain that leads to more accurate estimates compared to random sampling from the input space, which is a previously used alternative that can result in some missed linear regions and hence in inaccurate estimates about the information capacity of the neural network. This insight about sampling, allows then the authors to reduce the time to estimate the linear regions of special types of neural networks that exhibit some symmetry. This essentially reduces the number of samples needed and they experimentally verify their results.

Finally, the authors release OSCAR an open source library.

**Strengths:**

-connections of neural networks expressibity with tropical geometry, though they have been exploited in the past, are strengthened in this paper

-the paper presents a nice story that leads to faster sampling methods, both in theory and in practice.

**Weaknesses:**

-the main weakness I see in the paper is that, though well-motivated and interesting, it lacks technical depth. For example, there is essentially one main result stated as Theorem 4.3, and some intermediate results stated as Lemma 3.3 and Proposition 3.4. On the one hand, the latter two are simple observations about Puiseux polynomials, and on the other hand the proof of the Theorem 4.3 is not more than 3 lines (as shown in Appendix C). As such, I believe it is a nice transfer of ideas from tropical geometry to neural networks, but given that the connection was already there and used in prior works more than 10 years back, I don't think the better sampling algorithm is solid enough.

**Questions:**

-Would it be possible perhaps to strengthen the paper by proving depth-width separation results analogous to Telgarsky "Benefits of Depth in Neural Networks" using your techniques?

-Is there some notion of optimality associated with your sampling algorithm? Could it be further improved?

**Limitations:**

See weaknesses.

---

> ### Author Rebuttal · Authors · 2024-08-06
>
> We are grateful to the reviewer for their feedback on our work. We are happy that the reviewer found the soundness and contributions of our submission to be good; and that the reviewer appreciated the practicalities of our work in relation to existing theory at the intersection of tropical geometry and machine learning.
>
> We now respond to the questions and concerns raised by the reviewer.
>
> ### Technical Depth
>
> Our work lies at the intersection of pure mathematics and computer science and we are pleased that all reviewers found our presentation of the background and our contributions accessible and clear. As the reviewer pointed out among the strengths of our work, indeed, a major goal of our work was to adapt an existing theoretical connection to the practical setting to strengthen the current level of understanding of information capacity of neural networks, which we believe that we have achieved. We would like to
> further elaborate on several aspects concerning the technical depth of our contributions.
>
> The conciseness of our proofs should not be mistaken for a lack of technical depth, nor should it imply that our results lack utility. We strengthen existing theory by introducing powerful ideas into the established intersection of tropical geometry and neural networks. Two explicit examples are connecting the Hoffman constant and machine learning inductive biases to tropical geometry, which we have explicitly studied with symmetric group actions.
>
> The technical depth and impact of our work is further demonstrated by its implications and applications. Our geometric characterization of linear regions, for example, not only advances theoretical understanding but also has practical implications for improving computational efficiency in neural network analysis. The integration of our methods with the OSCAR system broadens the scope of symbolic computation in neural network research. In particular, the tools we provide can be used to analyze the tropical expressions of concrete neural networks; it also paves the way for the application of other computational tools and theory from the rich field of tropical geometry. For instance, this allows for the computation of some standard measures of complexity, such as the number of non-redundant monomials.
>
> We hope this clarifies the depth and significance of our contributions. We are confident that our work meets the high standards of technical rigor and innovation expected in the field, and we are grateful for the opportunity to further elaborate on these points.
>
> ### Depth–Width Separation
> The behavior of depth–width separation is not within the scope of interest of our work, so we did not explore this question of depth–width separation theoretically, though upon further reflection on this relationship in order to respond to the reviewer’s question on this relationship, we conclude that our work may be able to offer some empirical interpretation of the depth–width separation, since the number of non-redundant monomials gives a concrete limit on what functions may be represented. Empirically, from the perspective of our methods, it is 'easier' to increase width than depth, since increasing the latter can lead to a significant increase in the number of monomials that appear and thus to more complicated functions. We computed the number of monomials (after removing redundant ones from the numerator and the denominator) in the tropical expression of random neural networks with the same number of hidden units. As expected, the deeper networks have more monomials than the shallow ones; see Table 2 attached.
>
> ### Optimality of Algorithms
> For our sampling algorithm, we con confirm its optimality in two aspects:
> 1. Sampling from the fundamental domain (Algorithm 2 of our original submission) gives an estimation of the true value of number of linear regions. We further have tightness of the upper bound and lower bound of the sampling ratio (Theorem 4.3 of our original submission, line 242). In fact, in experiments, we always obtain a 100% sampling ratio (Figure 1 of our original submission), which means the sampling algorithm outputs the true value.
> 2. The computation of Hoffman constant gives an estimation of the smallest sampling radius (Proposition 3.4 of our original submission, line 174), which further gives the smallest (optimal) sampling box that hits all linear regions. In other words, it gives the optimal region for our Algorithm 4 to be theoretically correct.
>
> We thank the reviewer once again for the helpful feedback provided and
> hope that we were able to adequately address all concerns.

---

> ### Comment · Reviewer_vnGc · 2024-08-11
> **post-rebuttal reviewer comment**
>
> The reviewer appreciates the author's response. The reviewer has read their comments and reread parts of the paper but still maintains the score unchanged. The reviewer would rewrite parts of the paper to highlight the novelty of the proofs, the proofs themselves and how they were significantly different from prior works, and finally the author's suggested connections to depth-width tradeoffs.

---

> > ### Author Response · Authors · 2024-08-12
> >
> > Thank you very much for your thoughtful response and for taking the time to revisit our paper in light of our comments.
> > In our revision, we will ensure that the novelty of our proofs, their distinction from prior works, and the connections to depth-width tradeoffs are further emphasized.
> >
> > One of the key contributions of our paper is demonstrating the computational feasibility of using the tropical form of a neural network to gain deeper insights into the geometry of linear regions and the complexity of the resulting tropical expression. **We noticed that the review has not yet mentioned this aspect**, and we would greatly appreciate any feedback you could provide on our symbolic computations for linear regions and tropical forms, as **we consider this to be a central component of our work**.
> > Thank you once again for your time and constructive feedback.

---

### Official Review · Reviewer_oTxQ · 2024-07-16

**Soundness:** 2
**Presentation:** 4
**Contribution:** 3
**Rating:** 4
**Confidence:** 4

**Summary:**

This work provides a geometric characterization of the linear regions in a neural network via the input space. Although linear regions are usually estimated by randomly sampling from the input space, stochasticity may cause some linear regions of a neural network to be missed. This paper proposes an effective sampling domain as a ball of radius R and computes bounds for the radius R based on a combinatorial invariant known as the Hoffman constant, which gives a geometric characterization for the linear regions of a neural network. Further, the paper exploits geometric insight into the linear regions of a neural network to gain dramatic computational efficiency when networks exhibit invariance under symmetry. Lastly, the paper provides code for converting trained and untrained neural networks into algebraic symbolic objects, useful for precisely the kinds of analysis this paper performs.

**Strengths:**

1. The authors present an interesting and novel way to analyze the capacity of a neural network using fundamental notions from tropical geometry.

2. The paper and theory were very clearly presented. In terms of writing, the presentation of the relevant tropical geometry for purposes of Hoffman constant estimation Section 3 was excellent.

**Weaknesses:**

1. My greatest concern in this paper stems from the experiments for upper and lower bound estimation for Hoffman constants given in Tables 1, 2, and 3 in the appendix. It seems the experimental upper and lower bounds computed there do not actually bound the true Hoffman constant. I understand that the upper bound may be loose due to the way it is estimated, but the lower bound should always be below the true Hoffman constant, as per my understanding. Yet for, say, the first of eight computations in table 1, the lower bound $H_L$ is $0.5460$, which is clearly above the true value of $H$, given to be $0.3298$. For that example, the upper bound $H^U$ is given to be $0.2081$, which is clearly not above the true value. This pattern continues, and the lower and upper bounds for the Hoffman constants seem to fluctuate somewhat arbitrarily around the true Hoffman constants, which is concerning. I am currently assuming that there is some kind of mistake with these experimental values and would like a clarification from the authors regarding this.

2. Due to the curse of dimensionality, this method for estimating the expressiveness of neural networks can only be applied to simple neural networks in practice. This is very apparent due to the way the numerical approach requires sampling on a mesh grid in an $n$-dimensional box (but is also true for the symbolic approach, that relies on the computation of the Puiseux rational function associated with a neural network, which becomes increasingly quite hard in higher dimensions). To the credit of the authors, they are up-front about this limitation, but this does significantly hinder the applicability of the presented results.

**Questions:**

My questions to the authors are listed below:

1. Why do the computed upper and lower bounds for the Hoffman constants in Tables 1, 2, and 3, seem wrong? Is there a mistake with this computation, and if so, can you provide the correct tables so that I may judge how tight/loose the bounds are? I elaborated on this in the first weakness of the "Weaknesses" section above.

### Additional Comments and Minor Corrections

The writing in this paper is quite good and the material is very clearly introduced. Nonetheless, I would like to give a non-exhaustive list of minor corrections below:

L103, 105: Please use \citet for in-text citations. This happens several times, but I will only mention it once.

L349: "to interpret and analyzed" -> "to interpret and analyze"

**Limitations:**

The authors adequately discuss the limitations of their work in Section 6.

---

> ### Author Rebuttal · Authors · 2024-08-06
>
> We would like to thank the reviewer for their careful and thoughtful reading
> of our work. We were pleased to read that the reviewer found our work
> novel and interesting, and that we were able to communicate and present
> the concepts and our contributions clearly. We especially appreciate that
> important issues on existing literature was raised via the reviewer’s questions,
> to which we now respond.
>
> ### Aim and Proof of Concept
>
> Firstly, we would like to clarify that the primary aim of our paper was
> to present a proof of concept rather than to develop highly optimized code
> for large-scale stress tests. Our focus was on establishing the theoretical
> foundations and demonstrating the feasibility of our approach. Consequently,
> we made no significant effort to optimize certain parts of the code for perfor-
> mance; this choice was made in order to demonstrate the direct applicability
> of established symbolic computation techniques to study neural networks.
> For example, the tropical Puiseux polynomial library, we utilized a direct
> implementation and relatively straightforward algorithms. We believe that
> there is substantial potential for improvement in this area, and more efficient
> algorithms could significantly enhance the performance of our approach,
> which we mentioned in the Discussion section of our submission for future
> research directions.
>
> ### Scalability and Higer-Dimensional Inputs
> Secondly, our experiments primarily focused on smaller networks to
> facilitate clear and comprehensible analysis. Nevertheless, the code we
> developed is capable of handling higher-dimensional cases than those presented
> in our experiments. For instance, our code can process networks with input
> dimension 28 × 28 (MNIST dimension) in roughly 1 minute. This capability
> demonstrates that our approach is in fact scalable, contrary to what our
> illustrative experiments might have suggested. In our revision, we plan to
> include these larger scale results alongside the smaller scale experiments,
> though we believe that the smaller scale ones provide a clearer and more
> readily accessible proof of concept in the sense that small networks allow us
> to more easily illustrate the key insights and mechanisms of our approach.
>
> We do appreciate the importance of these concerns and hope that we
> have addressed them adequately. We do agree with the reviewer that with
> further optimization and development, the techniques we proposed could be
> applied to more complex and larger-scale neural networks.
>
> ### Hoffman Constants
>
> We are very grateful to the reviewer for their careful reading of our work
> and especially for pointing out the mistakes in Table 1–3 concerning the
> computed Hoffman constants. Indeed, the lower bound should always be
> below the true Hoffman constants. Upon further careful investigations of the
> public code we used to compute the true Hoffman constants due to Peña et
> al. (2018), we find that the reviewer was indeed correct that the values
> we reported were erroneous because the public code does not always return
> correct value. To complete our experiments, we then tested examples without
> using the code of Peña et al. (2018), and instead implemented a brute
> force computation to find the maxima over all submatrices. We have provided
> updated tables for these experiments and these new results show that the
> lower bounds are always below true values, which is consistent with the
> theory.
>
> With regard to upper bounds, Theorem 3.9 (line 214) in our original
> submission shows that the Hoffman constant of a matrix $A$ is bounded by
> the inverse of the smallest nonzero singular value of $A_J$ , where $J$ ranges over
> all rows of $A$. In our implementations, to determine a nonzero value, we
> set the threshold to be $10^{−10}$. However, it seems that a value of $10^{−10}$ is so
> strict that in fact many small singular values are ruled out. After revisiting
> our computations and listing all the singular values, we found that many
> small singular values have magnitude between $10^{−30} ∼ 10^{−14}$. Thus the
> singular values indeed provide upper bounds, in practice, so the theory is
> again consistent. In practice, however, in agreement with the reviewer, we
> also acknowledge that the upper bounds can be tightened and in the revision,
> we will mention this as a noteworthy direction of future research. We believe
> that such a study could also have positive implications on the Stewart–Todd
> condition measure of a matrix.
>
> ### Minor Comments and Corrections
> We appreciate the few typographical and formatting errors flagged by the
> reviewer and will commit to careful editing and proof-reading in our revision.
> Thank you once again for your valuable feedback.
>
> **References:** (cited in our original submission)
>
> Javier F. Peña, Juan C. Vera, and Luis F. Zuluaga. *An algorithm to compute the Hoffman constant of a system of linear constraints.* arXiv preprint
> arXiv:1905.06366 (2019).

---

> > ### Comment · Reviewer_oTxQ · 2024-08-11
> > **Rebuttal Response**
> >
> > I would like to thank the authors for providing a rebuttal. My concerns regarding the Hoffman constants have been somewhat assuaged, but I am surprised that the code the authors were using from prior work was wrong and that this was caught only during the review process. Out of an abundance of caution, I would like to maintain my reject rating. I believe this work has some merit, but greater care needs to be taken when computing bounds. I hope that the authors will improve their work and produce a higher quality resubmission.

---

> > > ### Author Response · Authors · 2024-08-13
> > >
> > > Thank you for taking the time to carefully review our work and for providing your thoughtful feedback. We appreciate your acknowledgment of the improvements we’ve made in addressing the concerns regarding the Hoffman constants. While we respect your decision to maintain your initial rating, we believe it is important to note that despite satisfactorily addressing the concerns raised, this has not influenced the final evaluation. This process and outcome diminishes the impact that constructive dialogue is intended to have on the review process.
> > > This being said, we have found this process to be an invaluable opportunity to further refine our research, and we remain committed to making any necessary improvements. We sincerely thank you again for your constructive comments and the time invested in reviewing our work and engaging in discussion.

---

### Author Rebuttal · Authors · 2024-08-06

We would first and foremost like to thank all the reviewers for their time invested in reading and providing thoughtful feedback on our work. We were pleased to find that they found the work well-written and clearly presented, and that they found the intersection of tropical geometry with neural networks to be interesting and insightful in assessing network information capacity.

We appreciate that our contribution straddles quite different areas of mathematics and computer science, thus, we would like to take this opportunity to reiterate the implications of our work, and especially to emphasize their novelty and highlight their importance. Existing work by Zhang et al. (2018) establishes the equivalence between feedforward neural networks with ReLU activation and tropical rational functions; this theoretical connection was then used to give an analytic upper bound on the number of linear regions of a network. The number of linear regions of a neural network is an important quantity that has been well-studied in the machine learning literature that
provides a measure of the network’s information capacity. Our work builds on the earlier work of Zhang et al. (2018) to adapt these concepts to practicality, for instance, where the number of linear regions is expected to evolve during training. One of our main contributions is an efficient numerical method to compute this quantity and thus better understand how training affects the information capacity of a neural network. Furthermore, by considering not just the number of linear regions but also their geometric characteristics that can be described algebraically (as in the spirit of inductive biases in geometric deep learning), we are able to further increase the computational efficiency of our numerical approach to understanding information capacity.

A question that was common to several reviewers asks about the scalability and curse of dimensionality of our work. We understand these these concerns pertain to the applicability of our contributions, given that our numerical experiments were carried out on rather small networks. Our reasoning for these proof-of-concept-scaled experiments was to highlight our third contribution, which is a practical connection between neural networks and not just tropical geometry, but also the vast existing literature and codebases on symbolic computation in computational algebraic and polyhedral geometry, in the sense that, thanks to our theoretical contributions, these can be directly and immediately applied to study neural networks (specifically, OSCAR was used in our implementations). In response to the question on effective utility given the smaller scale of our experiments: Firstly, we acknowledge that there exist many computational improvements that can be implemented to the existing computer algebra software to better understand neural networks, such as parallelization and GPU execution; this is a promising and important line of future research (mentioned in the Discussion section of our submission). To the best of our knowledge, ours is the first work to connect neural networks to computer algebra software. Secondly, we would like to point out that our methods do indeed scale to the input dimensionalities of common and well-known benchmarks, such as MNIST. Additional experimental results are provided in Table 1 attached.

**References:** (cited in our original submission)

Liwen Zhang, Gregory Naitzat, and Lek-Heng Lim. *Tropical geometry
of deep neural networks.* In International Conference on Machine Learning
(2018), pages 5824–5832. PMLR (2018).

---

### Decision · Program_Chairs · 2024-09-25

**Decision:**

Reject

**Comment:**

The submission presents a tropical geometry perspective of neural networks aimed at identifying theoretical and practical frameworks to count linear regions and better understand their geometry.

Reviewers praised the clarity and novel consideration of certain concepts from tropical geometry in context of neural nets, found the technical content accessible and not unnecessarily convoluted, and the proofs and concepts clearly presented. However, concerns were raised in regard to the provided experiments, particularly the correctness of the estimated bounds; the computational complexity of the proposed methodology and thus limited applicability (also acknowledged in the submission); and that, although well motivated and interesting, the paper lacked technical depth. This latter concern extended to aspects related to rigour in tropical geometry and positioning of the work in the context of existing literature.

The authors contended that the primary aim of the article was to present a proof of concept and that they did not make any significant efforts to optimise certain parts, that experiments focused on smaller networks to facilitate clear analysis. They offered to include larger experiments in a revision as well as other improvements including additional references. They acknowledged that a table in their original submission had mistakes and that some computations had used thresholds producing incorrect results. Authors also contended that the conciseness of their proofs should not be mistaken for lack of technical depth, although their argumentation focused on other aspects, particularly implications and applications, and integration with OSCAR. After the rebuttal and discussions, the reviewers were not sufficiently persuaded to change their reject recommendations and offered suggestions how the article could be improved.

After carefully considering the reviews, responses, discussion, and also reading the article, I found that the article is written in a clear way and presents an interesting exposition of concepts like the Hoffman constant which appears promising. However, in my assessment, the scope of technical innovations and implications is limited and currently not sufficiently well developed to warrant publication at NeurIPS. This is in agreement with some of the opinions expressed by the reviewers. Therefore, I recommend to reject the submission. The discussion identified several points of possible improvement, particularly greater care with numerical experiments.